# Input and Output Privacy in Cross-Silo Federated Settings: an MPC+DP Approach

## Abstract

We address the problem of training a machine learning model on data held by multiple data holders in a cross-silo federated setup while ensuring privacy guarantees. Existing Federated Learning (FL) solutions with Differential Privacy (DP) or Secure Multiparty Computation (MPC) with DP are often limited to either horizontal or vertical partitioning and typically suffer from accuracy loss compared to a centralized setting. We propose an MPC-based approach for training differentially private linear models that supports any partitioning scenario and effectively combines MPC and DP. Our solution employs MPC protocols for both model training and output perturbation using Laplace-like noise. By simulating a trusted curator through MPC, our approach provides the benefits of global DP without requiring an actual trusted party. The resulting MPC+DP method achieves accuracy comparable to a centralized DP setup while maintaining privacy guarantees in a cross-silo federated setup.

## 1 Introduction

Machine Learning (ML) models are widely deployed in real-world applications. However, training accurate and effective ML models often requires access to large amounts of data. In many practical scenarios, relevant training data is fragmented across multiple organizations and remains siloed due to privacy concerns, regulatory constraints, and/or competitive advantage. This necessitates cross-silo federated learning approaches, where ML models can be trained collaboratively across multiple data holders (clients in federated learning) without compromising data privacy.

Furthermore, in a federated scenario, data can be distributed in various ways. One common scenario are *horizontally distributed settings* where each data holder has records (rows) of data with the same feature set (HFL). For example, hospitals treating COVID-19 patients could collaborate to build a model that predicts the length of hospital stay by combining their individual patient records. Another scenario are *vertically distributed settings*, where different data holders hold different features (columns) about the same set of individuals (VFL). An example of this is an ML model that relies on lab test results as well as healthcare bill payment information about patients, which are usually managed by different departments within a hospital system. Data can also be distributed in a mixed manner that fits neither the horizontal nor the vertical partitioning scheme. In the 2023 U.S. U.K. PETs Prize competition, for instance, participants were challenged to develop a federated learning solution for financial fraud detection with data from banks and from the Society for Worldwide Interbank Financial Telecommunication (SWIFT) (Vos et al. (2024)). In this scenario, SWIFT has data about transactions between customers across banks, while each bank has more detailed data about its own customers. An ideal federated learning setup should support any data distribution, enabling privacy-preserving analytics over pooled datasets. This allows collaboration for privacy-preserving model training across organizations and within different departments of the same organization.

The importance of enabling privacy-preserving model training in federated setups has spurred a large research effort in this domain, most notably in the development and use of Privacy-Enhancing Technologies (PETs), prominently including Federated Learning (FL) (Kairouz et al. (2021)), Differential Privacy (DP) (Dwork

et al. (2014)), Secure Multiparty Computation (MPC) (Cramer et al. (2015)), and Homomorphic Encryption (HE) (Lauter (2022)). Each of these techniques has its own (dis)advantages.

DP has become the gold standard for providing formal privacy guarantees in trained ML models, mitigating the risk of leaking sensitive training data – often referred to as "output privacy" (Ding et al. (2017); Apple Differential Privacy Team (2017); Hartmann & Kairouz (2023)). These privacy guarantees come at the cost of accuracy loss that is inversely proportional to the *privacy budget*. In centralized settings, where all data is available in one place, these trade-offs tend to be less severe, making DP more practical in such scenarios (global DP) (Ponomareva et al. (2023)).

Federated settings employ approaches based on (combinations of) FL, MPC, or HE, allowing data holders to train models without sharing their raw data, thereby ensuring "input privacy". But these approaches do not guarantee protection of the underlying datasets (such as against membership inference and reconstruction attacks) once the trained model is published, i.e. after federated training, and hence they are often combined with DP. Most studies in federated settings focus on the combination of FL and DP[1] and incur severe utility loss when compared to the centralized setting. Our research aims to mitigate this utility loss by combining MPC and DP in federated settings, effectively emulating global DP from the centralized setting. While existing studies have explored MPC-DP combinations (see Section 3.3), most focus specifically on horizontally federated learning (HFL). In contrast, our proposed approach works with any arbitrary data distribution in a federated setting.

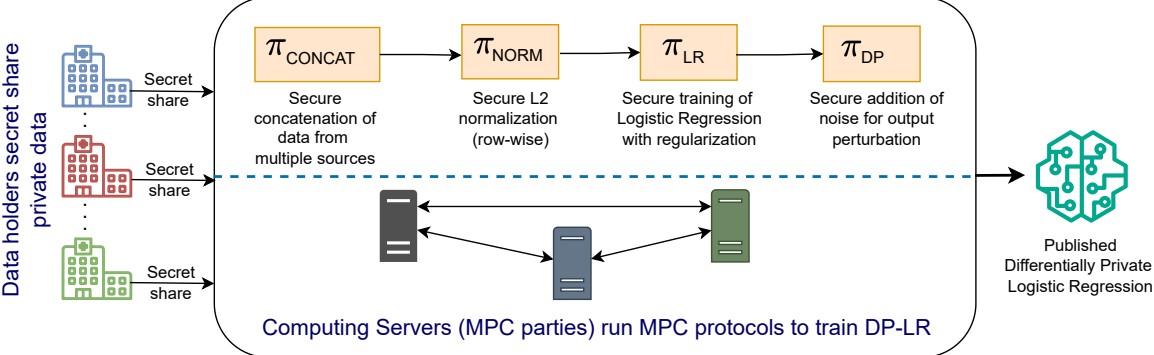

Figure 1: **Overview of our modular approach**: Data holders secret-share their private data with computing servers (illustrated here with 3 servers), which train an $\epsilon$-DP model using MPC. Noise is generated and added within MPC itself to ensure DP guarantees, without reliance on a trusted entity. **For linear models**: Step 1: $\pi_{\mathsf{CONCAT}}$ merges secret shares of data from all data holders to form secret shares of a single union dataset, thus mimicking a centralized setup. Step 2: $\pi_{\mathsf{NORM}}$ normalizes the secret-shared data, row-wise, to satisfy L2-norm requirements for output perturbation. Step 3: $\pi_{\mathsf{LR}}$ trains an LR model with regularization on normalized secret shares of data. Step 4: $\pi_{\mathsf{DP}}$ generates secret shares of noise and adds it to the secret shares of the weights of the trained model. Finally the DP model is published.

**Our Approach.** Unlike traditional FL, where each client trains local models on their respective datasets, our approach comprises of computing servers (MPC parties)[2] that run MPC protocols. These MPC protocols train ML models over secret shares of the combined dataset without requiring any entity to disclose its private information to anyone. This enables our approach to work with any data distribution. Our focus is on generalized linear models (Castro Torres & Akbaritabar (2024)). To ensure differential privacy guarantees, we adapt an output perturbation technique from the centralized setting (Chaudhuri et al. (2011))[3]. The MPC parties, after training the ML model, run MPC protocols to generate secret shares of the necessary noise and add it to the secret shares of weights of the trained model to satisfy DP requirements. We show that this procedure yields the same accuracy as in the global DP model. Indeed, the MPC protocols effectively play

---

[1]Either through local DP or distributed DP; see Section 3.1 in Ponomareva et al. (2023).

[2]MPC parties can be different from the data holders, as in an MPC-as-a-service scenario.

[3]For our datasets, our results show that output perturbation performs best; and is in general efficient for our scenario.

the role of a *trusted curator implementing global DP*. The resulting model can be published in the clear, or used for private inference on top of MPC. Below we briefly discuss the advantages of our approach.

Our use of MPC-as-a-Service setting, where external computing MPC servers (separate from the data holders a.k.a. clients in FL) perform secure computations using MPC protocols, offers several practical advantages. It enables participation from multiple data holders, including those without significant computational resources, thus addressing scalability and client selection bias common in traditional FL setups. Unlike traditional FL, our approach does not require clients to remain online throughout the training – once data is secret-shared, data holders can go offline. The MPC servers though are required to be online and synchronous. This is in fact a practical advantage of our MPC-as-a-Service setting, where MPC servers operate as robust, dedicated services, similar to other infrastructure components, and can be made fault-tolerant through standard redundancy and recovery mechanisms. Furthermore, because of the MPC-as-a-Service approach, the scalability of our method only depends on the data dimensionality and not on the number of clients. While MPC and DP can be combined in various ways, our approach is applicable to a broad class of DP mechanisms, including Laplace-based sampling methods that we consider in our work (which does not benefit from the distributional properties of Gaussian noise, as exploited in some prior works including Das et al.). Integrating DP into MPC protocols in this way requires careful protocol design for such mechanisms to ensure both privacy and utility. In our approach, noise is securely and independently sampled within the MPC framework, ensuring that neither the data nor the noise is exposed.

MPC-as-a-Service settings also introduces a few challenges. Establishing an MPC-as-a-Service infrastructure requires dedicated, non-colluding servers. Technical expertise is required in design and development of efficient and effective MPC protocols, including the correct integration of MPC protocols with DP mechanisms to ensure end-to-end privacy.

We summarize our contributions below.

- We propose a combination of MPC and DP to train linear models in a federated setting that yields ML utility similar to the centralized setting.
- Our proposed *modular* approach works for any arbitrary distributed data setting.
- We propose an MPC protocol to sample noise from a multidimensional power exponential distribution, generating Laplace-like noise. Unlike recent works that focus on DP-SGD like mechanisms (gradient or objective perturbation), our focus is on output perturbation.
- We demonstrate the effectiveness of our approach on real-world datasets. Our solution obtained the highest accuracy in the iDASH2021 Track III competition on confidential computing, where the challenge was to propose a federated learning algorithm for training of a model to predict the risk of wild-type transthyretin amyloid cardiomyopathy using medical claims data from different hospitals, while providing DP guarantees[4].

## 2 Preliminaries

### 2.1 Differential Privacy

DP is concerned with providing aggregate information about a dataset $D$ without disclosing information about specific individuals in $D$ (Dwork et al. (2014)). A dataset $D'$ that differs in a single entry from $D$ is called a neighboring database. A randomized algorithm $\mathcal{A}$ is called $(\epsilon, \delta)$-DP if for all pairs of neighboring databases $D$ and $D'$, and for all subsets $S$ of $\mathcal{A}$'s range,

$$P(\mathcal{A}(D) \in S) \leq e^\epsilon \cdot P(\mathcal{A}(D') \in S) + \delta. \tag{1}$$

In other words, $\mathcal{A}$ is DP if $\mathcal{A}$ generates similar probability distributions over outputs on neighboring datasets $D$ and $D'$. The parameter $\epsilon \geq 0$ denotes the *privacy budget* or privacy loss, while $\delta \geq 0$ denotes the probability of violation of privacy, with smaller values indicating stronger privacy guarantees in both cases. $\epsilon$-DP is a shorthand for $(\epsilon, 0)$-DP. $\mathcal{A}$ can for instance be an algorithm that takes as input a dataset $D$ of training examples and outputs an ML model. An $(\epsilon, \delta)$-DP randomized algorithm $\mathcal{A}$ is commonly created out of an

---

[4]`http://www.humangenomeprivacy.org/2021/competition-tasks.html`

algorithm $\mathcal{A}^*$ by adding noise that is proportional to the sensitivity of $\mathcal{A}^*$. We describe the noise generation technique that we use to this end in detail in Section 4.

There are three major properties of DP that help in designing DP algorithms. *Sequential composition* states that if $m$ differentially private algorithms $\mathcal{A}_1, \ldots, \mathcal{A}_m$ are applied to the same dataset, where each $\mathcal{A}_i$ satisfies $\epsilon_i$-DP, then the combination of all $m$ outputs satisfies $(\sum_{i=1}^{m} \epsilon_i)$-DP. This means that privacy loss accumulates linearly with each additional access to the data. *Parallel composition* states that if disjoint subsets of a dataset are each processed using DP algorithms $\mathcal{A}_1, \ldots, \mathcal{A}_m$, where each $\mathcal{A}_i$ satisfies $\epsilon_i$-DP, then the overall privacy guarantee is determined by the $\max(\epsilon_i)$ used among them. That is, if each disjoint subset is analyzed with an $\epsilon$-DP algorithm, the total algorithm still satisfies $\epsilon$-DP. This allows more efficient use of the privacy budget when data is partitioned as disjoint subsets. The *post-processing property* of DP guarantees that if $\mathcal{A}$ is $\epsilon$-DP, then $g \circ \mathcal{A}$ is also $\epsilon$-DP where $g$ is an arbitrary function. In other words, any arbitrary computations performed on DP output preserves DP without any effect on the privacy budget $\epsilon$.

## 2.2 Secure Multiparty Computation

MPC is an umbrella term for cryptographic approaches that allow two or more parties to jointly compute a specified output from their private information in a distributed fashion, without revealing the private information to each other (Cramer et al. (2015)). MPC is concerned with the protocol execution coming under attack by an adversary which may corrupt one or more of the parties to learn private information or cause the result of the computation to be incorrect. MPC protocols are designed to prevent such attacks being successful, and can be mathematically proven to guarantee privacy and correctness. We follow the standard definition of the Universal Composability (UC) framework (Canetti (2000)), in which the security of protocols is analyzed by comparing a real world with an ideal world. For details, see Evans et al. (2018). Below, we describe the standard threat models used in the MPC literature which we adhere to in this paper.

An adversary can corrupt a certain number of parties. In a *dishonest-majority* setting the adversary is able to corrupt half of the parties or more if he wants, while in an *honest-majority* setting, more than half of the parties are always honest (not corrupted). Furthermore, the adversary can have different levels of adversarial power. In the *semi-honest* model, even corrupted parties follow the instructions of the protocol, but the adversary attempts to learn private information from the internal state of the corrupted parties and the messages that they receive. MPC protocols that are secure against semi-honest or *"passive"* adversaries prevent such leakage of information. In the *malicious* adversarial model, the corrupted parties can arbitrarily deviate from the protocol specification. Providing security in the presence of malicious or *"active"* adversaries, i.e. ensuring that no such adversarial attack can succeed, comes at a higher computational cost than in the passive case. The protocols in Section 4 are sufficiently generic to be used in dishonest-majority as well as honest-majority settings, with passive or active adversaries. This is achieved by changing the underlying MPC scheme to align with the desired security setting.

As an illustration, we describe the well-known additive secret-sharing scheme for dishonest-majority 2PC with passive adversaries. In Section 5 we additionally present results for honest-majority 3PC and 4PC schemes with passive and active adversaries; for details about those MPC schemes we refer to Araki et al. (2016); Dalskov et al. (2021). In the additive secret-sharing 2PC scheme there are two computing parties, nicknamed *Alice* and *Bob*. All computations are done on integers, modulo an integer $q$. The modulo $q$ is a hyperparameter that defines the algebraic structure in which the computations are done. A value $x$ in $\mathbb{Z}_q = \{0, 1, \ldots, q-1\}$ is secret shared between Alice and Bob by picking uniformly random values $x_1, x_2 \in \mathbb{Z}_q$ such that $x_1 + x_2 = x \mod q$. $x_1$ and $x_2$ are additive shares of $x$ (which are delivered to Alice and Bob, respectively). Note that no information about the secret value $x$ is recovered by any of the individual shares $x_1$ or $x_2$, but the secret-shared value $x$ can be trivially revealed by combining both shares $x_1$ and $x_2$. The parties Alice and Bob can jointly perform computations on numbers by performing computations on their own shares, without the parties learning the values of the numbers themselves.

For protocols in the passive-security setting, we use $[\![x]\!]$ as a shorthand for a secret sharing of $x$, i.e. $[\![x]\!] = (x_1, x_2)$. Given secret-shared values $[\![x]\!] = (x_1, x_2)$ and $[\![y]\!] = (y_1, y_2)$, and a constant $c$, Alice and Bob can jointly perform the following operations, each by doing only local computations on their own shares[5]:

---

[5]We often omit the modular notation for conciseness.

- Addition of a constant ($z = x + c$): Alice and Bob compute $(x_1 + c, x_2)$. Note that Alice adds $c$ to her share $x_1$, while Bob keeps the same share $x_2$. This operation is denoted by $[\![z]\!] \leftarrow [\![x]\!] + c$.

- Addition ($z = x + y$): Alice and Bob compute $(x_1 + y_1, x_2 + y_2)$ by adding their local shares of $x$ and $y$. This operation is denoted by $[\![z]\!] \leftarrow [\![x]\!] + [\![y]\!]$.

- Multiplication by a constant ($z = c \cdot x$): Alice and Bob compute $(c \cdot x_1, c \cdot x_2)$ by multiplying their local shares of $x$ by $c$. This operation is denoted by $[\![z]\!] \leftarrow c[\![x]\!]$.

Multiplication of secret-shared values $[\![x]\!]$ and $[\![y]\!]$ is done using a so-called *multiplication triple* (Beaver (1992)), which is a triple of secret-shared values $[\![u]\!]$, $[\![v]\!]$, $[\![w]\!]$, such that $u$ and $v$ are uniformly random values in $\mathbb{Z}_q$ and $w = u \cdot v$. Given that they have a multiplication triple, Alice and Bob can compute $[\![d]\!] = [\![x]\!] - [\![u]\!]$ and $[\![e]\!] = [\![y]\!] - [\![v]\!]$, and, in a communication step, *open* $d$ and $e$ by disclosing their respective shares of $d$ and $e$ to each other. Next, they can compute $[\![z]\!] = [\![w]\!] + d \cdot [\![v]\!] + e \cdot [\![u]\!] + d \cdot e$, which is equal to $[\![x \cdot y]\!]$. We denote this operation by $[\![z]\!] \leftarrow \pi_{\mathsf{MUL}}([\![x]\!], [\![y]\!])$. Each multiplication requires a fresh multiplication triple. Such triples can be predistributed by a trusted initializer (TI). In case a TI is not available or desirable, Alice and Bob can simulate the role of the TI, at the cost of additional pre-processing time and computational assumptions, see Mohassel & Zhang (2017).

Building on these cryptographic primitives, MPC protocols for other operations can be developed, including for privacy-preserving training of ML models and noise generation to provide DP guarantees (see Section 4). Our protocols use well known subprotocols for division $\pi_{\mathsf{DIV}}$ of secret-shared values, square root $\pi_{\mathsf{SQRT}}$ of secret-shared values, and generation of random values from a uniform distribution $\pi_{\mathsf{GR-RANDOM}}$ (Keller (2020)).

## 3 Related Work

Our approach preserves input privacy, i.e., it ensures that the training datasets are not exposed (except under $\epsilon$-DP guarantees) to anyone but their original data holders during (1) model training and (2) publication or inference. As we describe below, existing methods either do not fully protect input privacy, or they do so at the cost of higher accuracy loss than our approach.

### 3.1 MPC/HE based Model Training

Many cryptography based methods have been proposed for privacy-preserving learning of ML models with data from multiple data holders such as Zhang et al. (2020); So et al. (2022). These include training for linear regression models (Gascón et al. (2017); Agarwal et al. (2019)), (ensembles of) decision trees (Lindell & Pinkas (2000); de Hoogh et al. (2014); Abspoel et al. (2021); Adams et al. (2022)), and neural network architectures (Mohassel & Zhang (2017); Wagh et al. (2019); Guo et al. (2020); De Cock et al. (2021)).

These techniques protect input privacy during training while still, in principle, producing the same ML models that one would obtain in the clear (i.e. when no encryption is used). The latter is both a blessing, as there is no accuracy loss, and a problem, as upon model publication or during inference, the trained models leak the same kind of information as models trained in-the-clear (Fredrikson et al. (2015); Tramèr et al. (2016); Song et al. (2017); Carlini et al. (2019)). Because these methods do not provide DP guarantees and are therefore prone to membership inference attacks Zhao et al. (2025), we do not compare with them in Section 5.

### 3.2 DP and FL based Model Training

Much of the literature on training DP models (Abadi et al. (2016)) is developed for the *global* DP (a.k.a. *central* DP) paradigm, which assumes the existence of a trusted curator (aggregator) who collects all the data and then trains a DP model over it, e.g. by adding noise to the gradients or the model coefficients. These methods do not preserve input privacy, since data holders need to disclose their datasets to the aggregator. In a *local* DP approach, privacy loss is controlled by having the data holders add noise to their input data or local models *before* disclosing it to the aggregator such as in Liu et al. (2024) who employ local DP using

DP-SGD on the client side; which results in substantial utility degradation (Row 4 in Table 4 provides an upper estimate of the accuracy that can be achieved in this way). We eliminate the need for a trusted curator by simulating this entity through MPC protocols that are run either directly by the parties themselves or as in a MPC-as-a-service model[6].

Another related existing approach combines Federated Learning (FL) with DP. In FL, each of the data holders participates in model training on their end and only exchanges trained model parameters or gradients with the central server (Kairouz et al. (2021)). A widely adopted method to provide DP guarantees in a FL setting is through local DP, where the data holders can add noise to protect the values that they send to the central server. In Section 5 we compare with such an approach in which the data holders perturb their model coefficients before sending them to the central server for aggregation. Another alternative to addressing the utility loss associated with local DP is distributed DP, which primarily works with Gaussian-like mechanisms, whereas our approach relies on variants of the Laplace mechanism. Moreover, these approaches work only in the horizontally distributed datasetting, while our approach (see Section 4) works in the vertically distributed setting as well.

### 3.3 Combinations of MPC and DP

The key idea in our proposed approach is to train DP models while performing as much of the computations as possible in MPC protocols in order to preserve accuracy. MPC and DP for ML have been well studied in isolation, but the strong privacy protections that can result from their synergy are still being explored (Wagh et al. (2021); Das et al. (2025)). We combine MPC and DP to protect training data privacy during training *and* during inference. In practice, we simulate the trusted curator present in the centralized DP model by using MPC. While in the past such an approach was avoided, due to the high computational cost of training the models on top of MPC, we argue that, with advances in protocols and computing power, the higher utility that can be obtained in this way justifies its adoption in several situations. The idea to replace the trusted curator from the global DP paradigm with MPC to get better privacy at the same high utility is gaining traction. Böhler & Kerschbaum (2021) for instance have explored this idea for detecting the top $k$ most frequent items across different datasets. They let each party locally compute partial noises which are then combined, which is different from our approach of letting the parties execute an MPC protocol to jointly sample secret-shared noise.

Combining MPC with DP has been proposed in the context of FL, where the data is either *horizontally* distributed (see e.g. Maddock et al. (2024); Acar et al. (2017); Jayaraman et al. (2018); Pathak et al. (2010)) or *vertically* distributed (see e.g. Tajeddine et al. (2020)). Existing solutions use cryptographic protocols (not necessarily MPC) and DP, such as in Pathak et al. (2010); Chase et al. (2017); Jayaraman et al. (2018); Byrd & Polychroniadou (2020); Truex et al. (2019), in order to train individual models on the datasets held by the data holders and aggregate these models by averaging their coefficients. These approaches are designed for horizontally partitioned data. Gu et al. (2021) propose a framework for combining MPC and FL, but it only works for the case of horizontally partitioned data. Moreover, each (MPC) server in Gu et al. (2021) generates noise using the Gaussian mechanism locally, secret shares and then adds to the gradient updates – adopting distributed DP which does not achieve accuracy similar to the centralized setup. Choquette-Choo et al. (2021) propose a framework that combines HE, MPC and DP inspired by PATE to collaboratively train a model in a way that is applicable only to a horizontal setting. All of the above mentioned solutions do *not* work for vertically partitioned data, unlike our method. Moreover, our solution trains the final model on the union of all the individual datasets, thus essentially obtaining the same utility that is achievable in the trusted curator scenario.

Very few recent solutions combine MPC and DP to work for any arbitrary partition, i.e., a single framework that works for horizontal, vertical, and mixed data distributions. Pentyala et al. (2024) do so for synthetic data generation rather than training DP linear models. While Das et al. (2025)'s solution could in principle work with any arbitrary distribution, their focus is on gradient perturbation, where each MPC party locally generates secret shares of samples from a Gaussian distribution. In contrast, our solution applies output

---

[6]MPC-as-a-sevice assumes an infrastructure setup where the MPC parties are the independent and non-colluding servers (controlled by different entities). This model allows for data from multiple data holders, i.e. the number of data holders can be different from the number of computing parties.

perturbation and is based on a Laplace-like mechanism[7]. To the best of our knowledge, no solutions exist for training $\epsilon$-DP linear models with output perturbation in arbitrarily partitioned scenarios that achieve accuracy at par with a centralized setup.

# 4 Method

## 4.1 Overview

We work in the scenario described in Fig. 1 distinguishing between the *data holders* who hold the training datasets, and the *computing servers* who run the MPC protocols for model training, noise generation, and noise addition. Our solution works in scenarios in which each data holder (e.g. hospital or bank) is also an MPC party (i.e. owns the computing server), as well as in scenarios where the data holders outsource the computations to untrusted servers (computing servers) instead (i.e. MPC-as-a-service scenario). Though we demonstrate our solution for 2, 3, and 4 computing servers, the MPC protocols we propose are generic in nature and so our solution works with any number of computing servers as well as data holders. This can be achieved by choosing an appropriate underlying MPC scheme for the desired number of computing servers (which are MPC parties) (see Section 2.2)

The data holders secret-share their data with a set of computing servers. In all MPC protocols used in this paper, secret sharings are in $\mathbb{Z}_q$ with $q = 2^\lambda$, i.e. a power 2 ring[8]. Since all computations in MPC are done over integers in $\mathbb{Z}_q$ (see Section 2.2), the data holders first convert the real numbers in their data to integers using a fixed-point representation (Catrina & Saxena (2010)) and subsequently split the integer values into secret shares which are sent to the computing servers (see Fig. 1). While the original value of a secret-shared number can be trivially revealed by combining the shares, the secret-sharing based MPC schemes ensure that nothing about the inputs is revealed to any subset of the computing servers that can be corrupted by an adversary. This means, in particular, that no server by itself learns anything about the actual values of the inputs.

Next, the computing servers proceed by performing computations on the secret shares. The servers run MPC protocols that output an ML model protected by DP. The resulting model can be used for private inference (on top of the underlying MPC protocol) or made open to the public because it is protected with $\epsilon$-DP guarantees and preserves privacy. In particular, the computing servers:

1. Jointly run $\pi_{\mathsf{CONCAT}}$ to merge the distributed data.

2. Jointly run MPC protocol $\pi_{\mathsf{LR}}$ to L2 normalize the training data, and to subsequently infer an LR model using L2 regularization from the normalized data. At the end of this protocol, the coefficients of the model are secret-shared between the parties.

3. Jointly run MPC protocol $\pi_{\mathsf{DP}}$ to add a noise vector to the secret-shared model coefficients. At the end of this protocol, the noisy coefficients of the model are secret shared between the parties.

4. Disclose their shares of the LR coefficients so that they can be combined in a final $\epsilon$-DP LR model.

As the noise in step 3 is generated and added to the weights using MPC, the computing parties will not learn it, hence they will not be able to retrieve the actual model coefficients from the noisy coefficients that are disclosed in step 4.

The core of our solution is the MPC protocol $\pi_{\mathsf{DP}}$ that implements a mechanism for providing $\epsilon$-DP. It does so by perturbing the coefficients of a trained logistic regression (LR) model with the addition of a noise vector $\eta$ that is sampled according to the density function

$$h(\eta) \propto e^{-\frac{n\epsilon\Lambda}{2}\|\eta\|} \tag{2}$$

---

[7]Das et al. (2025) follows the concept of distributed DP for Gaussian distributions and does not require joint noise sampling, whereas our approach relies on a Laplace-like mechanism necessitating joint sampling to provide pure DP.

[8]In Section 5 we present results with $\lambda = 64$ for a varying number of data holders, and for 2, 3, and 4 computing parties.

In the above expression, $n$ is the number of instances that were used to train the LR model, and $\Lambda$ is the regularization strength parameter used during training. This technique provides $\epsilon$-DP provided that **(C1)** each input sample (i.e., each row's feature vector) has an L2 norm of at most 1; and **(C2)** the LR model is trained using L2 regularization. If **(C1)** and **(C2)** are fulfilled, then the sensitivity of LR with regularization parameter $\Lambda$ is at most $\frac{2}{n\Lambda}$ (Chaudhuri & Monteleoni (2008); Chaudhuri et al. (2011)).

Note that our proposed method is modular in nature. For example, $\pi_{\mathsf{CONCAT}}$ can be an MPC protocol that simply concatenates and unions the datasets, or runs a secure alignment protocol before securely joining them (Wang et al. (2024))[9]. At the end of $\pi_{\mathsf{CONCAT}}$, the computing servers hold secret shares of the merged training examples $[\![S]\!]$.

## 4.2 Protocol $\pi_{\mathsf{LR}}$ for Model Training

At the beginning of the LR training protocol, the computing servers have secret shares of a set of labeled training examples $S = \{([\![\mathbf{x}]\!], [\![t]\!])\}$, each consisting of a secret-shared input feature vector $\mathbf{x}$ of length $m$ and a secret shared label $t$. $\pi_{\mathsf{LR}}$ is based on an existing MPC protocol for training a LR with full gradient descent (GD) (Keller (2020)). We extended this protocol in two ways. First, to satisfy condition **(C1)**, before the start of model training, we let the computing parties apply L2 normalization to the secret shares of each training example $[\![\mathbf{x^{norm}}]\!]$ by running $\pi_{\mathsf{NORM}}$. Pseudocode for $\pi_{\mathsf{NORM}}$ is provided separately in Prot. 1 because we also need it as a subprotocol for $\pi_{\mathsf{DP}}$. If the data is horizontally distributed across the data holders, then each data holder can apply sample-wise L2 normalization to their own instances before secret sharing the training instances with the computing servers. The computing servers in this case can skip the use of $\pi_{\mathsf{NORM}}$ for this purpose, which will reduce the training runtime. Second, to comply with condition **(C2)**, we implemented regularization by changing the weight update rule to $[\![\Delta\mathbf{w}]\!] \leftarrow C[\![\Delta\mathbf{w}]\!] - \alpha[\![\Delta\mathbf{w}]\!] - \Lambda\alpha[\![\mathbf{w}]\!]$. In this expression, $[\![\mathbf{w}]\!]$ and $[\![\Delta\mathbf{w}]\!]$ are the weights and gradients as maintained in secret-shared form throughout the model training; $C$ is the momentum; $\alpha$ is the learning rate; and $\Lambda$ is the regularization penalty. Pseudocode for $\pi_{\mathsf{LR}}$ is provided in Appendix A.

---

**Protocol 1:** $\pi_{\mathsf{NORM}}$ for secure $L2$ normalization

**Input** : A secret-shared vector $[\![\mathbf{x}]\!]$ of length $d$
**Output**: Secret-shared L2 normalized vector $[\![\mathbf{x^{norm}}]\!]$

**1** Declare vector $[\![\mathbf{x^{norm}}]\!]$ of length $d$
**2** $[\![S]\!] \leftarrow 0$
**3 for** $i = 1$ *to* $d$ **do**
**4**     $[\![S]\!] \leftarrow [\![S]\!] + \pi_{\mathsf{MUL}}([\![x_i]\!], [\![x_i]\!])$
**5 end**
**6** $[\![v]\!] = \pi_{\mathsf{DIV}}(1, \pi_{\mathsf{SQRT}}([\![S]\!]))$
**7 for** $i = 1$ *to* $d$ **do**
**8**     $[\![x_i^{norm}]\!] \leftarrow \pi_{\mathsf{MUL}}([\![x_i]\!], [\![v]\!])$
**9 end**
**10 return** $[\![\mathbf{x^{norm}}]\!]$

---

## 4.3 Protocol $\pi_{\mathsf{DP}}$ for Noise Generation

At the end of MPC protocol $\pi_{\mathsf{LR}}$, the coefficients $\mathbf{w}$ of the trained LR model are secret-shared between the servers. Next, the servers run the MPC protocol $\pi_{\mathsf{DP}}$, presented in pseudocode in Prot. 2, to generate noise and add it to the model coefficients to provide DP guarantees. Protocol $\pi_{\mathsf{DP}}$ implements the output perturbation method (or sensitivity method) (Chaudhuri & Monteleoni (2008); Chaudhuri et al. (2011))

---

[9]In Sec. 5, for simplicity we assume that the records are already aligned in case of vertical partitioning and the computing servers need to simply concatenate them using the right axis. This assumption does not affect the demonstration of our novel contributions, as the results remain valid and $\pi_{\mathsf{CONCAT}}$ itself is not a novel component.

while providing input privacy. While the original output perturbation method relies on the fact that the model coefficients are known or disclosed to a single entity, such as a trusted curator, we do not make such an assumption. Instead, the model coefficients remain secret-shared among the computing servers, neither of which knows the true values of the model coefficients. The challenge is for the computing servers to jointly generate noise that is appropriate for the true model coefficients that they cannot see, without learning the true value of the noise. Indeed, no entity should learn the true value of the noise, so that the noisy model coefficients can safely be disclosed at the end of the process (see step 4 in the overview at the beginning of this section), without leaking information that would violate the DP guarantees.

---

**Protocol 2:** $\pi_{\mathsf{DP}}$ for secure output perturbation

**Input** : A secret-shared vector $[\![\mathbf{w}]\!]$ with $d$ model coefficients $w_i$; regularization penalty $\Lambda$; total number $n$ of training examples; privacy budget $\epsilon$.
**Output:** Secret-shared vector $[\![\widetilde{\mathbf{w}}]\!]$ with perturbed model coefficients

**1** $[\![\mathbf{s}]\!] \leftarrow \pi_{\mathsf{GSS}}(d)$
**2** $[\![\mathbf{s}]\!] \leftarrow \pi_{\mathsf{NORM}}([\![\mathbf{s}]\!], d)$
**3** $[\![\gamma]\!] \leftarrow [\![0]\!]$
**4 for** $i = 1$ **to** $d$ **do**
**5**     $[\![u]\!] \leftarrow \pi_{\mathsf{GR-RANDOM}}(0, 1)$
**6**     $[\![u]\!] \leftarrow -\pi_{\mathsf{LN}}([\![u]\!])$
**7**     $[\![\gamma]\!] \leftarrow [\![\gamma]\!] + [\![u]\!]$
**8 end**
**9** $c \leftarrow 2/(n \cdot \epsilon \cdot \Lambda)$
**10** $[\![\gamma]\!] \leftarrow c[\![\gamma]\!]$
**11** Initialize vector $[\![\widetilde{\mathbf{w}}]\!]$ of length $d$ to $[\![\mathbf{0}]\!]$
**12 for** $i = 1$ **to** $d$ **do**
**13**     $[\![s_i]\!] \leftarrow \pi_{\mathsf{MUL}}([\![s_i]\!], [\![\gamma]\!])$
**14**     $[\![\widetilde{w}_i]\!] \leftarrow [\![w_i]\!] + [\![s_i]\!]$
**15 end**
**16 return** $[\![\widetilde{\mathbf{w}}]\!]$

---

In the output perturbation method, sensitivity is defined using the L2 norm, and the noise vector is sampled from a particular instance of a multidimensional power exponential distribution $h(\eta) \propto e^{-\frac{n\epsilon\Lambda}{2}\|\eta\|}$. Following Sánchez-Manzano et al. (2002), the computing servers can obtain secret shares of a vector $\mathbf{s}$ sampled according to the distribution $h(\eta)$, by following these steps, in which $d$ is the length of the vector (i.e. the number of model coefficients):

1. Generate a $d$-dimensional Gaussian vector $\mathbf{s}$. That is, each coordinate of the vector is sampled from a Gaussian distribution with mean zero and variance one. To this end, Line 1 in Prot. 2 calls $\pi_{\mathsf{GSS}}$ (see pseudocode in Prot. 3) which relies on the transform by Box & Muller (1958) to generate samples of the Gaussian unitary distribution, namely $\lceil d/2 \rceil$ pairs of Gaussian samples. For each pair, on Line 3–4 in Prot. 3, the computing servers securely generate secret shares of two random numbers $u$ and $v$ uniformly distributed in [0,1] by executing $\pi_{\mathsf{GR-RANDOM}}$. In $\pi_{\mathsf{GR-RANDOM}}$, each server generates $l$ random bits, where $l$ is the fractional precision of the power 2 ring representation of real numbers, and then the servers define the bitwise XOR of these $l$ bits as the binary representation of the random number jointly generated. On Line 5–8 in Prot. 3, the servers then jointly compute a secret sharing of $\sqrt{-2\ln(u)} \cdot \cos(2\pi v)$ and of $\sqrt{-2\ln(u)} \cdot \sin(2\pi v)$ using MPC protocols $\pi_{\mathsf{SQRT}}$, $\pi_{\mathsf{SIN}}$, $\pi_{\mathsf{COS}}$, and $\pi_{\mathsf{LN}}$ (Keller (2020)). In case $d$ is odd, one more sample needs to be generated. The servers do so on Line 11–12 in Prot. 3 by executing $\pi_{\mathsf{GSS}}$ to sample a vector of length 2 and only retain the first coordinate.

---

**Protocol 3:** $\pi_{\mathsf{GSS}}$ for secure sampling of a vector from a Gaussian distribution

---
**Input**    : Vector length $d$.
**Output**: A secret-shared vector $[\![\mathbf{s}]\!]$ of length $d$ sampled from Gaussian distribution with mean 0 and
            variance 1

**1** Declare vector $[\![\mathbf{s}]\!]$ of length $d$
**2 for** $i = 0$ **to** $d/2$ **do**
**3**     $[\![u]\!] \leftarrow \pi_{\mathsf{GR-RANDOM}}(0, 1)$
**4**     $[\![v]\!] \leftarrow \pi_{\mathsf{GR-RANDOM}}(0, 1)$
**5**     $[\![r]\!] \leftarrow \pi_{\mathsf{SQRT}}(-2\pi_{\mathsf{LN}}([\![u]\!]))$
**6**     $[\![\theta]\!] \leftarrow 2\pi[\![v]\!]$
**7**     $[\![s_{2i}]\!] \leftarrow \pi_{\mathsf{MUL}}([\![r]\!], \pi_{\mathsf{COS}}([\![\theta]\!]))$
**8**     $[\![x_{2i+1}]\!] \leftarrow \pi_{\mathsf{MUL}}([\![r]\!], \pi_{\mathsf{SIN}}([\![\theta]\!]))$
**9 end**
**10 if** $d$ *is odd* **then**
**11**     $[\![p]\!] \leftarrow \pi_{\mathsf{GSS}}(2)$
**12**     $[\![s_{d-1}]\!] \leftarrow [\![p_0]\!]$
**13 end**
**14 return** $[\![\mathbf{s}]\!]$

---

2. Normalize $\mathbf{s}$, that is divide each coordinate of $\mathbf{s}$ by its L2 norm (Line 2 in Prot. 2). After steps 1-2, the servers have secret-shares of a random $d$-dimensional vector on the unit sphere (this follows from the spherical symmetry of the multivariate Gaussian distribution).

3. In this step the computing servers change the magnitude of the vector obtained above to an appropriate value by sampling the gamma distribution $\Gamma(d, \frac{2}{n\epsilon\Lambda})$ to obtain a value $\gamma$, and multiplying each coordinate of the normalized vector produced in step 2 with $\gamma$. To generate a secret-shared sample $[\![\gamma]\!]$ from the $\Gamma(d, \frac{2}{n\epsilon\Lambda})$ distribution, on Line 3–8 in Prot. 2, the computing servers generate secret shares of $d$ independent samples from the exponential distribution with rate parameter one (here denoted by $\mathrm{Exp}(1)$) and add them. To generate secret shares of one such sample we use the inverse transform sampling over MPC, which consists of computing $-\ln u$, where $u$ is a random number with precision equal to $l$ bits generated by the computing servers within the interval $[0, 1]$:

   (a) On Line 5 the servers execute $\pi_{\mathsf{GR-RANDOM}}$ as in Prot. 3 to generate a random number with precision $l$ in $[0, 1]$. Denote this number by $u$.
   (b) On Line 6 the servers compute secret shares of $-\ln(u)$.

   Finally, on Line 9–11 the servers scale the sum by multiplying the secret shares with the factor $\frac{2}{n\epsilon\Lambda}$. On Line 13, they then multiply each coordinate of $\mathbf{s}$ with $\gamma$ to obtain the appropriate magnitude.

The obtained vector is then added to the vector of model coefficients on Line 14.

The importance of protocol $\pi_{\mathsf{DP}}$ stems from the fact that it enables the computing servers to generate secret shares of noise, without each server learning the true value of the noise that they add to the model coefficients in Line 14 of Prot. 2. The correctness of the protocol follows from the correctness of the inverse transform sampling algorithm, and the fact that $\mathrm{Exp}(1) = \Gamma(1, 1)$ and that $\sum_{i=1}^{d} \Gamma(1, 1) = \Gamma(d, 1)$. Moreover, it follows from the definition of the Gamma distribution that $c \cdot \Gamma(d, 1) = \Gamma(d, c)$. The security of the whole protocol follows from the security guarantees provided by the cryptographic primitives (Keller (2020)).

**Security and Privacy of** $\pi_{\mathsf{LR}} + \pi_{\mathsf{DP}}$**.** Below we formalize the security guarantees of $\pi_{\mathsf{LR}} + \pi_{\mathsf{DP}}$.

Input privacy: The underlying MPC schemes that we use in our method implement a secure arithmetic black-box MPC. They only perform operations over secret shares, and no information is leaked during the computation over the secret shares. Moreover, we use MPC sub-protocols $\pi_{\mathsf{LOG}}, \pi_{\mathsf{COS}}, \pi_{\mathsf{SIN}}, \pi_{\mathsf{GR-RANDOM}},$ and $\pi_{\mathsf{LR}}$ from MP-SPDZ. All these sub-protocols do not leak any information and are UC-secure. The novel

protocols that we propose ($\pi_{DP}$, $\pi_{GSS}$, and $\pi_{NORM}$) therefore do not leak any information to the MPC servers or the data holders, except for the perturbed weights of the LR model. Our protocol $\pi_{LR} + \pi_{DP}$, which is a composition of $\pi_{LR}$ and $\pi_{DP}$, UC-securely implements the ideal functionality $\mathcal{F}_{LR+DP}$ for privacy-preserving training of a differentially private logistic regression model.

---

**Ideal Functionality $\mathcal{F}_{LR+DP}$**

The functionality $\mathcal{F}_{LR+DP}$ operates as follows:

- Waits to receive input datasets $D_i$ from data holders and joins them to form $D$.

- Upon joining $D$, computes model weights $\mathbf{w} := \mathcal{A}^*(D)$ by training a logistic regression model $\mathcal{A}^*$ on $D$.

- Samples a noise vector $\mathbf{s} \sim (p(\eta) \propto h(\eta))$.

- Computes the privatized output $\tilde{\mathbf{w}} := \mathbf{w} + \mathbf{s}$.

- Reveals $\tilde{\mathbf{w}}$.

---

Output privacy: The fact that the resulting $\tilde{\mathbf{w}}$ provides $(\epsilon, 0)$-DP follows directly from the proof by Chaudhuri et al. (2011) regarding the privacy guarantees of the output perturbation method. Although in our approach the training and noise addition procedures are executed within MPC using fixed-point representations (in our case, precision of 32-bit), prior work by Mironov (2012) has shown that DP mechanisms remain valid under finite precision, even at 32 bits. Therefore, our protocol $\pi_{LR+DP}$, securely computes $\tilde{\mathbf{w}}$ and satisfies the same $(\epsilon, 0)$-DP guarantees.

*Remark:* An MPC+DP method for models like logistic regression, as we developed, is extremely valuable. First, while deep learning is state-of-the-art for many applications, GLMs remain a preferred method in domains such as finance and healthcare, where interpretability and regulatory compliance are essential. Second, logistic regression is often more practical and effective than deep learning in real world problems with data scarcity (e.g. training of AI models for rare diseases). The effectiveness of GLMs is supported, for instance, by our winning results on real-world datasets from the iDASH competition (Table 1) and Bailly et al. (2022) Third, and very relevant to our work, is that GLMs are significantly more efficient than deep learning. While it is technically possible to train larger neural networks in MPC Keller & Sun (2022), the computation and communication costs when doing so over encrypted data are orders of magnitude higher than for logistic regression. While there are applications where such high costs are justifiable to achieve higher accuracy, there are also many applications where logistic regression performs at par with neural networks, and training a logistic regression model in MPC is a far more economical and practical solution.

## 5 Results

### 5.1 iDASH Competition Results

We submitted our approach to a competition hosted by a National Center for Biomedical Computing funded by the NIH. In Track III of the iDASH 2021 competition, participants were invited to submit solutions for learning a ML model from training data hosted by two virtual centers, while providing DP guarantees. The centers represent data holders who have medical records of respectively 831 patients and 882 patients. Both datasets have the same schema, consisting of 1,874 boolean input attributes and a boolean target variable. The goal is to train a classifier for diagnosis of transthyretin amyloid cardiomyopathy using medical claims data (Huda et al. (2021)). Solutions submitted to the competition were required to run on two machines. They were evaluated in terms of (1) training runtime on two nodes with Intel Xeon E3-1280 v5 processors (4 physical cores, hyper-threading enabled) and 64 GiB memory; (2) accuracy on a held-out test of 429 patients.

Table 1 contains the results for the best performing teams satisfying the $\epsilon$-DP requirement (with $\epsilon$ set as 3 by the organizers). The first row corresponds to the approach presented in Section 4. We implemented the

$\pi_{\mathsf{LR}}$ and $\pi_{\mathsf{DP}}$ protocols in MP-SPDZ, an open source framework for MPC (Keller (2020))[10]. Being aware of the pitfalls of implementing DP with floating point arithmetic, our implementation follows the best practice of using fixed-point and integer arithmetic as recommended by, for example, OpenDP (The OpenDP Team (2020))[11]. As the underlying MPC scheme for the iDASH2021 competition, we used semi2k (a semi-honest adaptation of Cramer et al. (2018)) with mixed circuits that employ techniques using secret random bits (extended doubly-authenticated bits; edaBits) (Escudero et al. (2020)). This MPC scheme enables secure 2PC against semi-honest adversaries and complied with the requirements of the competition. As the regularizer for LR training, we used $N(\mathbf{w}) = \frac{1}{2}\mathbf{w} \cdot \mathbf{w}$, in which $\mathbf{w}$ denotes the vector of weights (coefficients) of the LR model, i.e. we used $\Lambda = 1$.

Table 1: Results for $\epsilon$-DP with $\epsilon = 3$ and data from two data holders, as provided by the iDASH2021 competition organizers

|    | Approach | PETs | Accuracy | Runtime |
|----|----------|------|----------|---------|
| 1. | $\pi_{\mathsf{LR}}$+$\pi_{\mathsf{DP}}$ (Section 4) | MPC & DP | 86.25% | $\sim$ 15,000 sec |
| 2. | feat. sel. and LR ensemble | DP | 85.31% | 31.942 sec |
| 3. | baseline (Section 5.1) | DP | 84.85% | 0.27 sec |
| 4. | decision tree based | DP | 84.38% | 0.09 sec |

All methods in Table 1 provide $\epsilon$-DP guarantees. The differences among the methods are in the utility (accuracy) and in the time taken to train a DP model. Our $\pi_{\mathsf{LR}}$ +$\pi_{\mathsf{DP}}$ approach achieved the highest accuracy of all methods, while taking the longest time to complete. Indeed, the runtime for the $\pi_{\mathsf{LR}}$ +$\pi_{\mathsf{DP}}$ approach is orders of magnitude larger than the runtimes for the other methods. This is because the $\pi_{\mathsf{LR}}$ +$\pi_{\mathsf{DP}}$ approach is the only method in Table 1 that uses MPC, while the other methods do not rely on cryptographic techniques. Approach 2 was based on feature selection and training an ensemble of LR models on selected feature subsets, while approach 4 was based on training a decision tree in a DP manner; these approaches were not created by us, and, to the best of our knowledge, their description has not been published in the open literature. In addition to the method from Section 4 we submitted an MPC-free baseline method to iDASH2021. We describe this method, which corresponds to approach 3 in Table 1, below as we also use it for further analysis and comparison in Section 5.2.1.

**Baseline Method.** The baseline technique follows a FL setup with horizontally distributed data in which each data holder locally trains a model on their data and adds noise to the model parameters at their end. Each data holder then shares its noisy parameters with a central server who performs averaging of the noisy model parameters and sends the result to the data holders. At the end of this process, each data holder holds the aggregated trained model. In more detail, in the baseline technique, each data holder:

1. Applies L2 normalization to its own instances;

2. Trains an LR model on its normalized instances[12];

3. Adds noise to the trained LR coefficients as per the output perturbation method (Chaudhuri et al. (2011)).

After going through steps 1-3, the data holders can each publish their perturbed LR coefficients, which we subsequently average to create a final model. Because steps 1–3 provide $\epsilon$-DP (Chaudhuri et al. (2011)), and since the datasets do not have common entries (a case of parallel composition), the overall solution provides $\epsilon$-DP due to the post-processing property of differential privacy.

---

[10]See `https://anonymous.4open.science/r/IDASH-MPCheavy-6D69/` for our code.
[11]See Appendix B for more details.
[12]We used the LR implementation from sklearn for this with penalty='l2' (L2 regularization) and $C = 1$ (the inverse of $\Lambda$).

Table 2: 5-fold CV accuracy results for varying number of data holders for $\epsilon$-DP with $\epsilon = 1$.

| # data holders | horizontally distributed | | vertically distributed | |
| | baseline | $\pi_{\mathsf{LR}} + \pi_{\mathsf{DP}}$ | baseline | $\pi_{\mathsf{LR}} + \pi_{\mathsf{DP}}$ |
|---|---|---|---|---|
| 2 | 85.79% | 87.98% | − | 87.98% |
| 4 | 83.36% | 87.98% | − | 87.98% |
| 8 | 76.92% | 87.98% | − | 87.98% |

### 5.2 Utility

#### 5.2.1 Horizontally and Vertically Distributed Data

For the results in Table 2 we distributed the data evenly among different numbers of data holders, both horizontally and vertically. We assume that the record alignment for vertical partitioning is already done using privacy preserving techniques as in Mohassel et al. (2020) prior to the start of the training. The baseline technique is only applicable when the data is horizontally distributed, while the $\pi_{\mathsf{LR}} + \pi_{\mathsf{DP}}$ approach works in the vertically distributed scenario as well. Even in the horizontally distributed scenario, the $\pi_{\mathsf{LR}} + \pi_{\mathsf{DP}}$ approach is preferable because it yields a higher accuracy, which becomes even more evident when the data is distributed among multiple data holders. The accuracy of the $\pi_{\mathsf{LR}} + \pi_{\mathsf{DP}}$ approach is independent of the number of data holders and the partitioning of data, as regardless of the partitioning, the computing servers still train a model over all the training data with $\pi_{\mathsf{LR}}$ and subsequently add noise *once* to the globally trained model coefficients with $\pi_{\mathsf{DP}}$, effectively simulating the global DP paradigm but without the involvement of a trusted curator. The baseline technique on the other hand adheres to the local DP paradigm in which each data holder adds noise to its local model, resulting in more noise in the final aggregated model. Furthermore, the utility of the $\pi_{\mathsf{LR}} + \pi_{\mathsf{DP}}$ approach is independent of the number of instances and/or features owned by each individual data holder, while the accuracy of the baseline technique degrades when individual data holders do not have sufficient instances to train local models that generalize well. This is especially relevant in biomedical applications that are characterized by high-dimensional datasets with relatively few instances.

#### 5.2.2 Effect of Privacy Budget $\epsilon$ on Accuracy of Models Trained with $\pi_{\mathsf{LR}} + \pi_{\mathsf{DP}}$

Table 3 shows the effect of the privacy budget $\epsilon$ on the accuracy of models trained with the $\pi_{\mathsf{LR}} + \pi_{\mathsf{DP}}$ approach. The accuracy is measured with 5-fold CV and over 3 iterations over the train and test data from Section 5.2.1. The training is done for 1000 epochs and $\Lambda = 1$. The results are as expected, with a larger privacy budget – i.e. less stringent privacy requirements – yielding more accurate models.

Table 3: Accuracy of models trained with $\pi_{\mathsf{LR}} + \pi_{\mathsf{DP}}$ for different values of $\epsilon$

| $\epsilon$ | 0.01 | 0.01 | 0.1 | 0.5 | 1 | INF |
|---|---|---|---|---|---|---|
| AVERAGED OVER 5 FOLDS | 54.81% | 80.66% | 62.57% | 89.18% | 89.39% | 89.47% |

#### 5.2.3 Comparison with Other Perturbation Techniques

For the $\pi_{\mathsf{LR}} + \pi_{\mathsf{DP}}$ approach (Section 4) and the baseline technique (Section 5.1), we adopted the sensitivity method that perturbs the model coefficients, i.e. the output perturbation method that was proposed as Algorithm 1 in Chaudhuri et al. (2011). In Table 4 we compare the output perturbation technique with other perturbation techniques, namely objective perturbation and gradient perturbation. For objective perturbation, we ran experiments with Algorithm 2 from Chaudhuri et al. (2011) that adds noise to the objective function itself[13]. For gradient perturbation, we ran experiments with DP-SGD (Abadi et al. (2016)) that adds noise

---

[13]We implemented this approach using IBM's Diffprivlib library
`https://github.com/IBM/differential-privacy-library`.

Table 4: Accuracy results obtained with 5-fold CV for $\epsilon$-DP with $\epsilon = 1$ and 2 data holders

|  | PERTURBATION | ACCURACY | |
|---|---|---|---|
| OUR APPROACH | OUTPUT | $\pi_{\mathsf{LR}} + \pi_{\mathsf{DP}}$ (SECTION 4) | 87.98% |
|  |  | BASELINE (SECTION 5.1) | 85.79% |
| OTHER APPROACHES | OBJECTIVE | BASELINE-OP | 49.40% |
|  | GRADIENT | BASELINE-DPSGD | 69.77% |
|  | SAMPLE-WISE DP | RANDOMIZED RESPONSE | 50% |

to the gradients[14]. For DP-SGD, we computed the required noise multiplier for given $\epsilon = 1, \delta = 1e - 5$, batch size of 1, 300 epochs, and the number of training examples each data holder holds. This was then passed as an argument to DP-SDG optimizer along with a clipping threshold of 1, learning rate of 0.1, and number of micro batches equal to the batch size.

In the BASELINE-OP method in Table 4, each data holder trains a differentially private LR model locally by perturbing the objective function. The resultant coefficients of the local models are then averaged, resulting in a final DP model. The BASELINE-DPSGD method is entirely similar, but in this method each data holder trains a differentially private LR model by perturbing the gradients learned during training, i.e. with DP-SGD.

As can be seen in Table 4, contrary to what one would expect based on the analysis in Chaudhuri et al. (2011), the accuracy results with this objective function perturbation method were not good on the iDASH2021 data, and far worse than those with the output perturbation method. We attribute this to the high-dimensional nature of the iDASH2021 data (many features and relatively few instances) which is very different from the datasets used for evaluation in Chaudhuri et al. (2011). Similarly, the LR models trained with DP-SGD on the iDASH2021 data are significantly less accurate than those protected with output perturbation.

We also included a baseline SAMPLE-WISE DP to demonstrate the naive local DP approach. For the IDASH2021 competition dataset, since the features are binary, we used randomized response as a sample-wise DP mechanism. We studied this in a setting with two data holders and a central entity that receives perturbed data from the data holders and trains a LR model on it. We performed 5-fold cross-validation, conducting 10 runs of randomized response per fold. The total privacy budget of $\epsilon = 1$ was distributed across both samples and features resulting in a per-value privacy budget of $\epsilon_v = 3.9 \times 10^{-7}$ for each value (due to sequential composition). Our results show, that in the absence of MPC and with data holders applying sample-wise DP, the average accuracy of the model is 50% with $\sigma = 0.01$ (the lowest among all approaches and the same as random guessing).

### 5.2.4 Comparison with Other Methods on Horizontally Partitioned Data

We evaluate our MPC+DP approach and compare against existing literature (Pathak et al. (2010) and Jayaraman et al. (2018)) that adopts a combination of PETs to train LR models and provide DP guarantees with the output perturbation technique[15]. The main distinction with our method, is that – similar as in the BASELINE method we adopted in Section 5 – these existing approaches let each data holder train a model locally and then add noise to the averaged model parameters using MPC+DP techniques. Because each data holder is required to train a model locally, these existing methods only work in scenarios where the data is horizontally partitioned, unlike our method which is suitable for vertically partitioned scenarios as well. We also note that the amount of noise added by each technique is different.

For the results in Table 5 we distributed the data evenly among different numbers of data holders, in a horizontal manner. We report 5-fold CV accuracy results averaged for 100 runs of noise generation mechanism

---

[14]We implemented this approach using TF-Privacy
https://www.tensorflow.org/responsible_ai/privacy/.

[15]https://github.com/bargavj/distributedMachineLearning

Table 5: Accuracy results for output perturbation obtained with 5-fold CV for $\epsilon$-DP with $\epsilon = 1$ on horizontally partitioned data

| Data Holders | Privacy Technique | Accuracy |
|---|---|---|
| 2 | OUR APPROACH ($\pi_{LR} + \pi_{DP}$, SECTION 4) | 87.98% |
| | PATHAK ET AL. (2010) | 86.43% |
| | JAYARAMAN ET AL. (2018) - MPC GRAD P | 86.42% |
| 4 | OUR APPROACH ($\pi_{LR} + \pi_{DP}$, SECTION 4) | 87.98% |
| | PATHAK ET AL. (2010) | 85.02% |
| | JAYARAMAN ET AL. (2018) - MPC GRAD P | 85.10% |
| 8 | OUR APPROACH ($\pi_{LR} + \pi_{DP}$, SECTION 4) | 87.98% |
| | PATHAK ET AL. (2010) | 84.10% |
| | JAYARAMAN ET AL. (2018) - MPC GRAD P | 84.24% |

to consider the randomness in the noise generation. We observe that for 2 data holders, all the techniques have close performance in terms of accuracy. Similar as for the BASELINE method in Section 5, the accuracy of the models trained by existing methods drops with an increase in the number of data holders. This may be because in existing approach, LR models are trained locally by the data holders, while our approach benefits from training an LR model on the combined data and learns a more generalized model. Moreover, our techniques are independent of how the data is distributed among data holders, unlike the methods in Table 5 that work only for horizontally distributed data.

## 5.3 Computational Efficiency

As Table 6 shows, the number of computing servers, the corruption threshold, and respective MPC schemes do have a substantial effect on the training time. The experiments for Table 6 demonstrate that our protocols are generic and can be used for a range of threat models. These experiments were run with the same training data as in Table 1 on co-located F32s V2 Azure virtual machines each of which contains 32 cores, 64 GiB of memory, and network bandwidth of upto 14 Gb/s. Every computing party ran on a separate VM instance (connected with a Gigabit Ethernet network). The times reported include computing as well as communication times. The training was run for 1000 epochs. with $\epsilon = 1$, $\Lambda = 1$ and with edaBits for mixed circuit computations.

In the horizontally distributed case, the data holders can L2-normalize their instances locally while in the vertically partitioned case the computing servers need to run MPC protocol $\pi_{NORM}$; this accounts for the difference in runtime between the horizontal and vertical partitioning. As expected, the corruption threshold has the most effect on the run time. Protocols that are secure for an honest majority of players (the protocols presented in Araki et al. (2016), and Dalskov et al. (2021)) are much faster than protocols secure against a dishonest majority (Cramer et al. (2018)). For the same corruption threshold, protocols secure against passive adversaries are faster than protocols secure against active adversaries. The four party protocol proposed in Dalskov et al. (2021) manages to obtain good run times for the case of active adversaries by further reducing the corruption threshold to 25%, i.e. one player out of four can be corrupted by an adversary and the protocol is still secure.

Our results show that MPC implementations for honest majority in the case of realistic sized datasets for genetic studies (a few hundred patients, and a few thousand features) are practical. We can train such models and add DP guarantees on top of MPC in less than 1.3 min for the case of honest majority protocols with passive security. Even in the case of stronger adversarial models, the training can be finished in a few hours, which is still practical for many applications where the increased accuracy payoff is valuable, especially with data that is distributed across multiple holders (Table 2).

Table 6: Runtimes of $\pi_{\mathsf{LR}} + \pi_{\mathsf{DP}}$ for different number $r$ of computing servers

| $r$ | Security | Horizontally distributed | Vertically distributed | MPC scheme |
|---|---|---|---|---|
| 2 | Passive | 35687 sec | 38056.92 sec | Cramer et al. 2018 |
| 3 | Passive | 75.83 sec | 454.83 sec | Araki et al. 2016 |
| 3 | Active | 500.28 sec | 1649.07 sec | Dalskov et al.2021 |
| 4 | Active | 160.50 sec | 838.02 sec | Dalskov et al.2021 |

Table 7: Runtimes of $\pi_{\mathsf{LR}} + \pi_{\mathsf{DP}}$ for different number $r$ of computing servers

| MPC SCHEME | $r$ | RUNTIMES | COMM. OVERHEAD |
|---|---|---|---|
| GOYAL ET AL. (2021) (PASSIVE) | 3 | 954.91 SEC | 57922.70 MB |
| | 4 | 1022.16 SEC | 83667.90 MB |
| | 5 | 2725.58 SEC | 366679.00 MB |
| | 7 | 5064.27 SEC | 711226.33 MB |
| CRAMER ET AL. (2000) & CHIDA ET AL. (2018) (ACTIVE) | 3 | 21213.21 SEC | 5247186.89 MB |
| | 4 | 23244.34 SEC | 7248822.06 MB |
| | 5 | 68176.24 SEC | 25728263.60 MB |
| | 7 | 131391.00 SEC | 70080327.38 MB |

### 5.3.1 Scalability of $\pi_{\mathsf{LR}} + \pi_{\mathsf{DP}}$.

The number of data holders in our solution is distinct from the number of computing servers. Our solution is general and works with any number of computing servers as well as data holders.

To study scalability of our approach, we generated synthetic datasets of varying dimensions as per Bailly et al. (2022). We varied both the number of rows (#samples ranging from 100 to 5,000,000) and columns (#features ranging from 100 to 10,000). The runtimes of our approach do not depend on the number of data holders, since our approach runs MPC protocols on the secret-shares of the joined dataset and the use of the MPC-as-a-Service model decouples scalability from the number of data holders. Figure 2 shows that the runtimes scale linearly with the total dataset dimensions (i.e., the size of $S$), which is in line with the literature. We ran our MPC protocols for 3PC passive threat model to train LR for 100 epochs with batch size of 32, $\epsilon = 1$ on 2.6 GHz 6-Core Intel Core i7 and latest version of MP-SDPZ. The results carry over to other threat models as per literature Keller & Sun (2022). The plots imply that our approach can be used in practice for smaller datasets and is feasible for larger datasets but at a high computational cost with strong privacy guarantees.

In Table 7, we report the runtimes and communication overheads to train an LR model with a varying number of computing servers $r$ ranging from 3 to 7. To have a comparison of runtimes and communication overhead for different values of $r$, we use the same MPC scheme for each security setting. The chosen MPC schemes can be used with any value of $r > 2$, and are different from the schemes that we use in Table 6 which were specific and efficient schemes for the given value of $r$. It is due to this use of different schemes that we observe a huge difference in runtimes when compared to the runtimes reported earlier. The schemes in Table 7 are run with secret sharings in $\mathbb{Z}_q$ where $q$ is a prime number[16] and with edaBits for mixed circuit computations.

The training was run on the complete training dataset from iDASH2021 consisting of 1713 training samples and 1874 features for 1000 epochs with GD, $\epsilon = 1$, and $\Lambda = 1$. The runtimes reported include computing as well as communication times. The total amount of data sent by all the computing parties is shown in the last column. The runtimes and the communication overhead increase with an increasing number of computing servers. This is because each server now needs to communicate with a higher number of parties servers, and the runtimes include communication times. Also, the active security settings take longer runtimes than their

---

[16]Defaults to None in MP-SPDZ and can be a maximum of bit length 256.

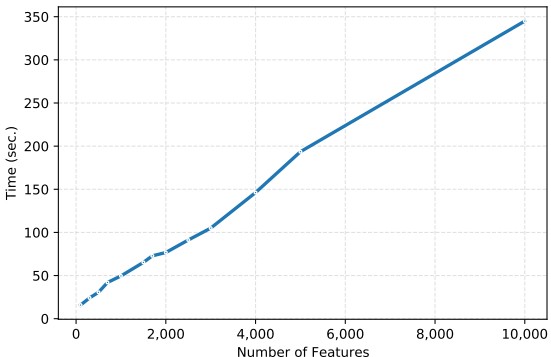

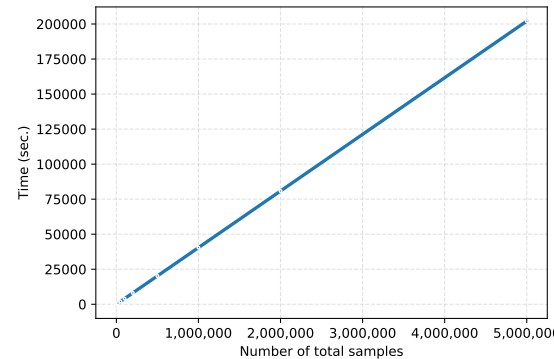

(a) Total number of samples is a constant of 200 with varying number of features from 100 to 10,000.

(b) Total number of features is a constant of 100 with varying number of samples from 100 to 5,000,000.

Figure 2: **Scalability Results**. The plots show runtime results for 3PC passive threat model with varying number of rows and columns of the combined dataset. Our approach is independent of number of data holders and depends on the dimensionality of the dataset only similar to centralized setting.

passive counterparts for a given $r$. These results are in line with the literature in MPC and demonstrate the our protocols are generic and work with different threat models and MPC schemes. The communication overhead in settings with a larger number of computing servers can be reduced with the use of a bulletin board functionality that enables efficient communication among many servers who are simultaneously involved in computations (Agarwal et al. (2019)).

### 5.4   Experiments on other datasets

We further evaluate our approach on the BC-TCGA and GSE2034 datasets of the iDASH 2019 competition[17]. Both datasets contain gene expression data from breast cancer patients which are normal tissue/non-recurrence samples (negative) or breast cancer tissue/recurrence tumor samples (positive) Xie et al. (2016). We perform experiments with a 5-fold CV, where the training data is distributed between 2 data holders in each fold.

**GSE2034**   Each instance in this train dataset is characterized by 12,634 continuous input attributes and a boolean target variable. There are 895 instances in total. In each iteration of the 5-fold CV, each data holder owns 447-448 instances, 20% of which is held out for testing.

**BC-TCGA**   Each instance in this train dataset is characterized by 17,814 continuous input attributes and a boolean target variable. There are 1,875 instances in total. In each iteration of the 5-fold CV, each data holder owns 937-938 instances, 20% of which are held out for testing.

The secure training is run for 20 epochs for the BC-TCGA dataset and 300 epochs for the GSE2034 dataset with $\Lambda = 1$ and $\epsilon = 1$. Table 8 shows accuracy results obtained with a 5-fold CV. To appreciate the inherent difference in difficulty between the GSE2034 and the BC-TCGA classification tasks, as the first row of results in Table 8 we include the accuracies obtained with a model trained in the central learning paradigm, i.e. when all the training data resides with a single data holder, and no noise is added to the model coefficients, i.e. $\epsilon = $INF. The other rows correspond to the federated setup from Section 5 with 2 data holders. The results are in line with the observation from Section 5 that the $\pi_{\mathsf{LR}} + \pi_{\mathsf{DP}}$ approach provides higher utility in general, when compared to other approaches. BASELINE-OP and BASELINE-DPSGD perform very poorly on these datasets. Pathak et al. (2010) and Jayaraman et al. (2018) perform similarly to our approach on GSE2034 dataset but our approach outperforms all baselines for BC-TCGA. The difference in relative performance between the datasets likely relates to the distribution of the data and feature characteristics between GSE2034 and BC-TCGA, where BC-TCGA seems to be an easier classification problem overall. The

---

[17]http://www.humangenomeprivacy.org/2019/competition-tasks.html

results in Table 8 results demonstrate that our method consistently performs same or better than existing baselines.

Table 8: Accuracy averaged over 5-fold CV with $\Lambda = 1$, $\epsilon = 1$

|  | GSE2034 | BC-TCGA |
|---|---|---|
| # INSTANCES $n$ | 895 | 1,875 |
| # FEATURES $d$ | 12,634 | 17,814 |
| CENTRAL LEARNING; 1 DATA HOLDER | 65.55% | 98.28% |
| CENTRAL LEARNING + DP; 1 DATA HOLDER | 64.70% | 95.83% |
| BASELINE (SECTION 5.1); 2 DATA HOLDERS | 51.92% | 91.37% |
| PATHAK ET AL. (2010); 2 DATA HOLDERS | 64.55% | 92.18% |
| JAYARAMAN ET AL. (2018); 2 DATA HOLDERS | 64.55% | 92.22% |
| BASELINE-OP; 2 DATA HOLDERS | 52.72% | 40.78% |
| BASELINE-DPSGD; 2 DATA HOLDERS | 47.87% | 67.10% |
| $\pi_{\mathsf{LR}}+\pi_{\mathsf{DP}}$ (SECTION 4); 2 DATA HOLDERS | 64.55% | 95.69% |
| RUNTIME FOR $\pi_{\mathsf{LR}}+\pi_{\mathsf{DP}}$; PASSIVE 3PC | 276.38 SEC | 57.30 SEC |

We additionally report the runtime to train the model using $\pi_{\mathsf{LR}} + \pi_{\mathsf{DP}}$ for these datasets to illustrate the variability in runtimes with respect to the number of training samples, epochs and a number of features in the dataset. We see an increase in runtimes for per epoch when compared to the runtimes per epoch on iDASH, which is attributed to a large number of features (about 10x of iDASH2021 for BC-TCGA and 7x for GSE2034). The runtimes for other threat models will follow a similar trend. We see that for larger datasets like these it is still practical to maintain the utility of the model while providing both input and output privacy guarantees.

## 6 Conclusion

We proposed a modular approach to train privacy-preserving linear models in a federated setting. To this end, we combined MPC and DP in a way that effectively offers the advantages of global DP but without the involvement of a trusted curator, as this curator is simulated by an MPC protocol instead. Our approach makes no assumptions about the data partitioning scenario, the number of computing parties or data holders, or the security setting in which it is applied. On the basis of linearity, $\pi_{\mathsf{LR}}$ is interchangeable with all linear learners without requiring reevaluation of noise variance. Our solution based on this approach led to 1st place in Track III of the iDASH 2021 Genome Privacy competition.

The trade-off between our MPC+DP approach that provides global DP and the baseline federated method with local DP can be summarized as operating cost (or running time) versus model accuracy. We empirically demonstrated the added utility of collaborative learning with MPC over the standard federated approach. The effect is particularly apparent as the number of disjoint collaborators grows. We also note that the baseline method as well as the existing methods that combine MPC with DP in FL, cannot be applied in cases where data is vertically partitioned which is a commonly-found scenario in medicine and advertising. In contrast, our MPC+DP method enables collaboration across a strictly larger space of applications.

Based on performance results, our protocol is extensible to larger datasets while remaining within a realistic time span for model training. It could be further improved through custom protocol implementations or the presence of a *correlated randomness dealer* in suitable scenarios. To further improve upon accuracy, a probable research direction is to introduce MPC protocols for feature selection Li et al. (2021) in both horizontal and vertical partitioning schemes.

**Broader Impact Statement**

Privacy-preserving machine learning is becoming increasingly important as organizations seek to leverage data while adhering to strict privacy regulations. Our research contributes towards this by proposing a modular and effective approach to collaborative training of $\epsilon$-DP models in arbitrarily partitioned data scenarios. Our research has impact in various domains, including healthcare, finance, and government analytics, where sensitive data from multiple silos must be analyzed as a whole without direct access to raw records; and where simple generalized linear models are effective and in use. While our method enhances privacy, it is important to consider potential trade-offs such as computational overhead and reduced accuracy due to DP, and potential attacks on implementations of DP. Users can replace the proposed protocols, as they are modular in nature, with much more efficient sub-protocols if available and explore optimizing these trade-offs to maximize the practical deployment.

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

## A   Pseudocode

Pseudocode for $\pi_{LR}$ is presented in Prot. 4. $\pi_{LR}$ is based on an existing MPC protocol for training a LR with gradient descent (Keller (2020)), which we extended in two ways to satisfy the conditions:

**(C1)** each input feature vector has an L2 norm of at most 1;

**(C2)** the LR model is trained using L2 regularization.

At the beginning of protocol $\pi_{LR}$, the computing parties have secret shares of a set of labeled training examples. To satisfy condition **(C1)**, on Line 1–3 the computing parties first apply L2 normalization to the secret shares of each training example by running protocol $\pi_{NORM}$; pseudocode for $\pi_{NORM}$ is provided separately in Prot. 1 in the paper.

The computing parties then begin secure training on the privately L2 normalized data from all the data holders. The training begins with initializing the secret shares of the weights (coefficients) of the LR model using Glorot uniform initializer (Glorot & Bengio (2010)). To this end, the computing parties execute protocol $\pi_{INIT}$ on Line 4. The training is carried out for $n_{iter}$ number of iterations (epochs), which is a public constant agreed upon by all computing parties along with the learning rate $\alpha$, the regularization penalty $\Lambda$, and the momentum $C$. In each epoch, the MP-SPDZ module $\pi_{FWD}$ for a secure forward pass is called on Line 6, followed by the MP-SPDZ module $\pi_{BKWD}$ for a backward pass on Line 7. The secret shares of the weights are then updated for every epoch using the MP-SPDZ module for updating the weights. We modified this module to satisfy **(C2)** with L2 regularization as per Line 8 in Prot. 4.

## B   Implementation of DP in MPC Protocols

It is well documented that implementing DP mechanisms using floating-point arithmetic can lead to catastrophic privacy compromises Mironov (2012). The most privacy-conscious choice, taken, for instance, by the OpenDP project[18] is to use fixed-point and integer arithmetic whenever possible. Our MPC protocols operate on fixed-point notation thus following the above paradigm. The accuracy of the model is another

---

[18]https://opendp.org/

---

**Protocol 4:** $\pi_{\mathsf{LR}}$ for secure logistic regression training

---

**Input** : A set $S = \{(\llbracket \mathbf{x} \rrbracket, \llbracket t \rrbracket)\}$ of secret-shared training examples, each consisting of a secret-shared input feature vector $\mathbf{x}$ of length $m$ and a secret shared label $t$; learning rate $\alpha$; regularization penalty $\Lambda$; momentum $C$; number of iterations $n_{iter}$.

**Output** : A secret-shared vector $\llbracket \mathbf{w} \rrbracket$ of weights $w_i$ that minimize the sum of squared errors over the training data

**1 for** *training examples* $(\llbracket \mathbf{x} \rrbracket, \llbracket t \rrbracket)$ *in* $S$ **do**
**2**     $\llbracket \mathbf{x} \rrbracket \leftarrow \pi_{\mathsf{NORM}}(\llbracket \mathbf{x} \rrbracket, m)$
**3 end**
**4** $\llbracket \mathbf{w} \rrbracket \leftarrow \pi_{\mathsf{INIT}}$                                        ▷ MP-SPDZ module for Glorot uniform initializer
**5 for** $i = 1$ **to** $n_{iter}$ **do**
**6**     Run $\pi_{\mathsf{FWD}}$                                                              ▷ MP-SPDZ module for forward pass
**7**     Run $\pi_{\mathsf{BKWD}}$                                                            ▷ MP-SPDZ module for backward pass
**8**     Run $\pi_{\mathsf{UPDATE}}$     ▷ Modified MP-SPDZ module for weight updates with the modified update rule for computing $\Delta \mathbf{w}$: $\llbracket \Delta \mathbf{w} \rrbracket \leftarrow C \llbracket \Delta \mathbf{w} \rrbracket - \alpha \llbracket \Delta \mathbf{w} \rrbracket - \Lambda \alpha \llbracket \mathbf{w} \rrbracket$
**9 end**
**10 return** $\llbracket \mathbf{w} \rrbracket$

---

point where the precision of weights could affect the overall result. Keeping this in mind, we used 32 bits of precision, which is more than sufficient to ensure the correct behavior of the training procedure.

We would like to stress that the finite precision issue is inherent to any implementation of DP on a digital computer – it is not specific to our work on DP implemented by MPC protocols. DP theory was created, for the most part, based on continuous distributions. However, all the practical libraries implement DP using finite precision arithmetic. That includes, for example, all the implementations of DP-SGD (which is based on the Gaussian mechanism). Das et al. (2025) adopt the discrete distributed Gaussian mechanism following the properties of a Gaussian distribution. Proposing a discrete version of the multidmensional power exponential distribution we use in this paper is research in itself. We further note that our proposed approach is modular in nature and $\pi_{\mathsf{GSS}}$ can be replaced by the appropriate sampling protocols for discrete distributions.

It is legitimate to wonder if security guarantees break down in the case when continuous DP mechanisms are implemented on digital computers. However, that question, which has to be asked of all implementations of DP mechanisms based on continuous distributions, is outside the scope of this paper.

