# OpenReview forum: "Input and Output Privacy in Cross-Silo Federated Settings: an MPC+DP Approach"
_TMLR — Rejected by TMLR_

### Review · Reviewer_Cbds · 2025-05-01

**Summary Of Contributions:**

This work presents an combination of Secure Multiparty Computation (MPC) and  Differential Privacy to perform a particular type of federated learning: the data holders outsource the computation to external federated learning computing servers.The MPC protocal ensures that the computing servers does not see the private data from data holders and the DP noise improves the privacy gurantee of the learned model.

**Audience:**

Yes

**Broader Impact Concerns:**

N/A.

**Claims And Evidence:**

Yes

**Requested Changes:**

1. Highlight the potential of the data holder and external computing servers setting in the introduction section.
2. Discuss the technical challenge in building a federated learning system for the new setting.
3. Discuss the difficulty in combining the MPC and DP algorithms.
4. Ablate the MPC part using sample-wise DP.

**Strengths And Weaknesses:**

**Strength**

1. This work considers an interesting and practical setting: the data holders may not hold any computing server. This setting can ease the cross-silo federated learning system setup and allow cross-silo federated learning being an easy-to-use service.

2. The methods are good fits for the problem. First, outsourcing the compute from data holders to external servers incurs privacy concerns. The authors use MPC to allow external computing servers to perform federated learning training using the protected data from data holders. Such an application makes sense. Then, the authors add DP noise to the learned federated learning model after the MPC step. The combination of MPC and DP seems new to me.

**Weakness**

1. This paper can be stronger if the authors compare their MPC approach against sample-wise DP approach. Using sample-wise DP, we may directly share the purturbed data without MPC.

2. In Section 4.1, condition (C1) requires each input feature vector has an L2 norm of at most 1. However, in Sectoin 4.2, the authors mention that "P. If the data is horizontally distributed across the data holders, then each data holder can apply sample-wise L2 normalization to their own instances". There is a gap between the feature-wise requiremnet and the sample-wise normalization.

---

> ### Author Response · Authors · 2025-05-17
> **Response addressing requested changes and weaknesses**
>
> Thank you for your positive feedback and suggestions. Please see our response to the requested changes below. We will include the following paragraphs 1-2-3 in our paper to address requested changes 1,2, and 3. Please let us know if it aligns with your expectations.
>
> 1. "We consider MPC-as-a-Service setting, where external computing MPC servers (separate from the data holders a.k.a. clients in FL) perform secure computations using MPC protocols. This separation offers several practical advantages. It enables participation from multiple data holders, including those without significant computational resources, thus addressing scalability and client selection bias common in traditional FL setups. Unlike traditional FL, our approach does not require clients to remain online throughout the training -- once data is secret-shared, data holders can go offline."
>
> 2. "This setting also introduces a few challenges. Establishing an MPC-as-a-Service infrastructure requires dedicated, non-colluding servers. Technical expertise is required in design and development of efficient and effective MPC protocols, including the correct integration of MPC protocols with DP mechanisms to ensure end-to-end privacy."
>
> 3. "While MPC and DP can be combined in various ways, our approach is applicable to a broad class of DP mechanisms, including Laplace-based sampling methods that we consider in our work (which does not benefit from the distributional properties of Gaussian noise, as exploited in some prior works including Das et al.). Integrating DP into MPC protocols in this way requires careful protocol design for such mechanisms to ensure both privacy and utility. In our approach, noise is securely and independently sampled within the MPC framework, ensuring that neither the data nor the noise is exposed."
>
> 4. We ran a new experiment to address the 1st identified weakness. For the IDASH2021 competition dataset, since the features are binary, we used randomized response as a sample-wise DP mechanism. We studied this in a setting with two data holders and a central entity that receives perturbed data from the data holders and trains a LR model on it. Similar to Table 4, we performed 5-fold cross-validation, conducting 10 runs of randomized response per fold. The total privacy budget of $\epsilon=1$ was distributed across both samples and features resulting in a per-value privacy budget of $\epsilon_v = 3.9 \times 10^{-7}$ for each value (due to sequential composition).
> Our results show, that in the absence of MPC and with data holders applying sample-wise DP, the average accuracy of the model is 50\%  with $\sigma=0.01$ (the lowest among all approaches and the same as random guessing). We will include these results in Table 4.
>
> 5. The 2nd identified weakness is based on a misunderstanding and is not an actual weakness. L2 normalization is applied per *row* (i.e. per sample/instance). In Section 4.1, "feature vector" refers to the vector of features for each row, not across all rows. Therefore, in Section 4.2, each data holder can independently apply sample-wise L2 normalization to their local data. We will rephrase this in Section 4.1 to avoid confusion as "(C1) Each input sample (i.e. each row's feature vector) has an L2 norm of at most 1."

---

> > ### Comment · Reviewer_Cbds · 2025-05-28
> > **Reply to Authors**
> >
> > Thanks for the update. I wonder if there is a better way to tune the sample-wise DP budget such that the accuracy does not degrade to 50%, which equals to the random guessing performance.

---

> > > ### Author Response · Authors · 2025-05-30
> > > **Response on sample-wise DP**
> > >
> > > Thank you for your helpful suggestion. In our earlier method for sample-wise DP, we assumed that features might be correlated and that samples may not be independent. As a result, we distributed the total privacy budget ($\epsilon = 1$) uniformly across both samples and features using sequential composition. For the IDASH dataset, it is reasonable to assume that the samples correspond to different individuals and are therefore independent. So now we applied parallel composition across samples (rows) and sequential composition across features. The per-feature privacy budget then becomes $\epsilon_v \approx 0.000533$. We re-ran the experiments and observed only a minimal improvement in performance, with the accuracy increasing slightly to 50.6\% with  $\sigma=0.0083$. We would be happy to experiment with additional ideas or mechanisms you might suggest for more effective application of sample-wise DP.

---

### Review · Reviewer_bUUn · 2025-05-01

**Summary Of Contributions:**

The authors introduce a method that combines MPC and DP to train models in a federated setting, using generalised linear models. The authors opt for the utilisation of output perturbation and ensuring input privacy and specifically offer functionality also for vertical FL. Their method benefits from using computing servers that run MPC protocols.

**Audience:**

No

**Claims And Evidence:**

No

**Requested Changes:**

- See weaknesses above
- A larger dataset would be interesting to see. How does the approach scale?
- I find the claim that the model at epsilon=1 outperforms the non-DP approach a bit weak. (“adding some noise can positively impact the generalization capability of the model.”) In my opinion, if a model performs on par with the non-DP setting on a dataset, this means that the dataset is very easy to solve.
- Figure 1 is more complicated that need be and could be more polished in my opinion
- How would this method work for more difficult ML methods?
- In Table 2, the introduced method out-performs the baselines. However, the argument is that the other methods suffer from the fact there might be data shortage on individual sights. Is there no other method available that would be a fairer comparison then? How are the results if all data holders have enough data to train their individual models?
- Are there more baselines that could be used? And all methods for all datasets to show a clear evaluation of performance in all aspects

Typos and minor changes:
- Page 6: “The data holders secret share their data with a set of computing server” should probably read “The data holders secretly share their data with a set of computing server”
- The paper reads quite lengthy, I think it would benefit from shortening
- Page 6: footnote 8 is weirdly placed in the text
- What does the footnote 3 mean in Table 1?

**Strengths And Weaknesses:**

Strengths:
- Vertical FL
- A setting for secure training over multiple data holders
- The code is publicly available
- The introduced method performs independently of the number of data holders (since it uses input privacy)

Weaknesses:
- The methodology is very old, using generalised linear models is not the most interesting ML approach
- The statement "and since the datasets do not have common entries (a case of parallel composition), the overall solution provides ϵ-DP due to the post-processing property of differential privacy" sounds like it is mixing up the concepts of parallel computing and post-processing? But I am no expert on this. Could you please elaborate on this?
- The runtime of the proposed method is very high
- How does this method scale to thousands of data holders and millions of rows?
- The overall writing is a bit over-complicated in my opinion. I think the paper would benefit from cuts and simplifications.
- What makes this method more performant - as claimed this is the only work that achieves performance on par with centralized settings? - - Where does this come from? Which (novel?) methodology ensures this?
- The authors claim that their method performs on par with centralised settings. *”We show that this
procedure yields the same accuracy as in the global DP model.“) Does that refer to the results in Table 8? I would not go as far as saying the results are on par. The centralised setting still out-performs the here introduced method. Which is fine, just I wouldn't state it like this.
- Would it be difficult to extend other existing methods to vertical FL? This should be a baseline as well.
- How do the other baseline methods perform on the datasets from Table 8?

---

> ### Author Response · Authors · 2025-05-20
> **Response for weaknesses 1,2 and 3.**
>
> Thank you for reviewing our work and providing valuable feedback. We spent time to address your concerns and answer your questions. Please see below for our responses, as much as possible in the order in which you raised questions.
>
> 1. **Relevance of GLMs.** While deep learning may seem more exciting for many ML researchers and is no doubt state-of-the-art in many applications, logistic regression is a tried and true method in practice.
> Firstly, GLMs remain a preferred method in domains such as finance and healthcare, where  interpretability and regulatory compliance are essential. Second, logistic regression is often more practical and effective than deep learning in real world problems with data scarcity (e.g. training of AI models for rare diseases). The effectiveness of GLMs is supported, for instance, by our winning results on real-world datasets from the iDASH competition (Table 1) and Bailly et al. (2022).
> Third, and very relevant to our work, is that GLMs are significantly more efficient than deep learning. While it is technically possible to train larger neural networks in MPC (Keller et al.), the computation and communication costs when doing so over encrypted data are orders of magnitude higher than for logistic regression. While there are applications where such high costs are justifiable to achieve higher accuracy, there are also many applications where logistic regression performs at par with neural networks, and training a logistic regression model in MPC is a far more economical and practical solution.
>
> **References**:
> - Bailly, Alexandre, et al. "Effects of dataset size and interactions on the prediction performance of logistic regression and deep learning models." Computer Methods and Programs in Biomedicine 213 (2022): 106504.
> - Keller, Marcel, and Ke Sun. "Secure quantized training for deep learning." International Conference on Machine Learning. PMLR, 2022.
>
> 2. **Clarity on description of DP properties.**
> ``Parallel composition", here, refers to the composition property of differential privacy (DP). In DP, when  disjoint datasets, say $D_1$ and $D_2$, are processed independently via $f_1$ and $f_2$ respectively, and each process satisfies $\epsilon$-DP, then the combined process (e.g., releasing results of $f_1(D_1)$ and $f_2(D_2)$) is $\epsilon$-DP. This is known as parallel composition. Furthermore, by the post-processing property of DP, any computation performed on these outputs (individually or jointly) does not increase the privacy loss, and thus the resulting outputs remain $\epsilon$-DP. We added a paragraph explaining the DP properties in the preliminaries section.
>
>    In our baseline method, each hospital holds the data of different patients (a case of horizontal partitioning) and independently trains DP LR models using output perturbation each satisfying $\epsilon$-DP. The parallel composition applies here, when hospitals publish their perturbed models, because the hospitals' datasets are disjoint. When these models are aggregated (i.e.~operations are performed on already DP outputs), the result (i.e. final DP LR model) remains $\epsilon$-DP due to the post-processing property of DP.
>
> 3. **On the higher runtime.** The substantially higher runtime of our approach when compared to traditional FL approaches is a direct result of the inherent tradeoff between privacy, utility, and computational cost. In traditional FL, the price for privacy is loss of utility. In MPC, the price for privacy is a high computational cost instead. Ensuring strong privacy guarantees (MPC and DP) and utility similar to the centralized paradigm with DP, introduces additional computational cost -- the price one pays for privacy and utility, which is desirable in many use cases such as in healthcare. Having said that, with advancements in hardware, and improvements and optimizations of MPC primitves, one can expect the runtimes to further reduce over time, making our approach a practical and viable solution. Moreover, unlike traditional FL, our approach can handle any form of data partitioning, including vertical and mixed.

---

> ### Author Response · Authors · 2025-05-20
> **Responses regarding performance**
>
> 4. **Regarding scalability, larger datasets, and more data holders.**  We conducted new scalability experiments using synthetic datasets generated as per Bailly et al. (2022), varying both the number of rows (\#samples ranging from 10k to 50,00,000) and columns (\#features ranging from 100 to 10,000). The runtimes of our approach do not depend on the number of data holders, since our approach runs MPC protocols on the secret-shares of the joined dataset (the total dataset size $S$ is constant) and the use of the MPC-as-a-Service model decouples scalability from the number of data holders. Our experiments show that the runtimes scale linearly with the total dataset dimensions (i.e. the size of $S$), which is in line with the literature.  We have included the scalability plots in the revised version of the paper.
> The runtimes though vary w.r.t. the number of MPC servers and the threat model as discussed in Section 5.3.1.
>
> 5. **Regarding performance.** The ability of our method to achieve utility on par with centralized settings stems from our core idea of using MPC to emulate a trusted centralized entity. What sets our approach apart is the novel integration of MPC and DP to enable global DP guarantees -- known to offer significantly better utility compared to local or distributed DP. While prior works have explored either MPC or DP in isolation, the literature on their combination is limited. Our contribution lies in performing the DP mechanism within the secure MPC computation, ensuring that noise is securely generated and applied without leaking any information. This design allows us to preserve strong privacy while maintaining high utility, though it does come with a computational cost, highlighting the trade-off between privacy, utility, and efficiency.
>
> 6. **Support for the claim that our method performs on par with centralized settings.** We confirm that our method (which gives input and output privacy) yields the same level of accuracy as in the centralized global DP model (which only gives output privacy).
> By "central learning; 1 data holder" in Table 8, we meant centralized training of LR but without DP (i.e. neither input nor output privacy). To give more clarity and to demonstrate our statement that our "procedure yields the same accuracy as in the global DP model", we have added a row in Table 8 for "central learning + DP; 1 data holder". Despite the fact that in our approach the data is split across 2 data holders, we get nearly the same accuracy as in the centralized + DP case where the data is controlled by only 1 data holder.
>
> 7. **Regarding baseline for vertical partitioning.** The baselines we evaluated in Table 4 and 5 (as discussed in Section 5.2.4) are not applicable to vertical FL, as each of these methods assumes that data holders can independently train local models, which in turn requires every data holder has access to all features and the corresponding labels of a sample. Training models locally and then aggregating them is much less straightforward when the data is vertically partitioned. If the reviewer has an idea for a vertical FL baseline method that would be good to add to the evaluation, we are open to implementing it and documenting the utility loss.
>
> 8. **"How do the other baseline methods perform on the datasets from Table 8?"** We have added 4 more rows with baseline method results to Table 8.
>
> 9. We agree with the reviewer that the comment that "adding some noise can positively impact the generalization capability of the model" was not a particularly solid explanation for what was observed in Table 3. The results in Table 3 were based on only one fold as mentioned in Section 5.2.2. This was not a good evaluation practice on our side. We reran the same experiment with 5-fold CV and over 3 iterations. The new results are as one would expect, with smaller privacy budgets consistently resulting in lower accuracy.
> | Epsilon Value | Accuracy (%) |
> |---------------|---------------|
> | 0.001         | 50.62         |
> | 0.01          | 54.81         |
> | 0.1           | 80.66         |
> | 0.5           | 89.18         |
> | 1           | 89.39        |
>     NO DP results in  89.47%
>
>     Given this, the performance for one fold that we originally had in the paper was very likely the result of random variation in that particular fold and not the dataset. We thank the reviewer for catching this, and we will update the paper accordingly.

---

> ### Author Response · Authors · 2025-05-20
> **Responses addressing remaining weaknesses**
>
> 10. **Extending approach to complex ML models.** Extending our approach to more complex ML models is feasible and aligns well with the modular design of our MPC-as-a-Service framework. It requires developing tailored MPC protocols for training specific ML models  for which many building blocks already exist in the literature (e.g., as demonstrated by Keller et al.). In addition, MPC protocols must be designed for the corresponding differential privacy (DP) mechanisms. This typically results in increased runtime, and the more complex the model, the higher the computational cost. To address this, one would need to carefully optimize the MPC protocols and identify operations that can be performed outside the MPC environment. For example, Das et al. leverage properties of the Gaussian mechanism to reduce the MPC burden while still providing DP guarantees.
>
> - Keller, Marcel, and Ke Sun. ``Secure quantized training for deep learning." International Conference on Machine Learning. PMLR, 2022.
>
> 11. **How are the results if all data holders have enough data to train their individual models?** If all data holders have enough data to train their individual models, then there is no reason to do federated learning. Our method is designed for scenarios in which individual data holders do not have enough data for  training accurate models -- such as in the case of rare diseases, with each hospital having data for only a few patients --, or when features are distributed vertically across institutions (e.g., hospitals or banks). In such settings, local models will under-perform due to either lack of data or incomplete feature sets.
>
> We will take care of the typos and suggested minor changes suggested. We also welcome concrete pointers on shortening the length of the paper (sections/parts that are not needed in the paper or could be shortened) and on improving the figure.

---

### Review · Reviewer_g5C2 · 2025-05-05

**Summary Of Contributions:**

This paper proposes an MPC-based federated learning framework that trains differentially private linear models across data holders in cross-silo settings, supporting both horizontal and vertical data partitioning. By simulating a trusted curator via secure multiparty computation (MPC) and applying Laplace-like noise for output perturbation, the method combines the strengths of MPC and global differential privacy (DP). It achieves accuracy comparable to centralized DP training while ensuring strong privacy guarantees without relying on a trusted third party.

**Audience:**

Yes

**Broader Impact Concerns:**

In this paper, the authors design a privacy-enhanced FL scheme for cross-silo data collaborative learning, aiming to achieve a better privacy-utility trade-off by integrating MPC and DP. Specifically, the proposed method can be effectively applied to data-sensitive fields such as healthcare, insurance, and finance. However, the proposed method seems to be only applicable to generalized linear models, which may limit its application scenarios because deep learning models generally achieve better performance.

**Claims And Evidence:**

Yes

**Requested Changes:**

This paper presents a framework designed to protect both input and output privacy in federated learning (FL) settings, leveraging techniques such as additive secret sharing and functional encryption. While the paper attempts to address important concerns in privacy-preserving FL, there are several critical weaknesses that limit its contribution and clarity.

1. Lack of a Clear Threat Model

One major shortcoming of this paper is the absence of a well-defined threat model. The authors do not explicitly state the attacker's goals, background knowledge, or capabilities. Without this, it is difficult to evaluate the robustness of the proposed privacy guarantees. For example, it is unclear whether the adversary is passive or active, internal or external, or whether collusion among clients or servers is considered. This lack of specification makes it hard to assess the relevance and completeness of the proposed mechanisms in realistic federated environments.

2. Absence of Formal Privacy or Security Proofs

The paper does not provide formal security proofs or theoretical guarantees to back the claimed privacy-preserving properties. While the authors describe using additive secret sharing and functional encryption, there is no rigorous proof of correctness or security (e.g., simulation-based privacy, indistinguishability, or semantic security) presented for the protocols. In the domain of privacy-preserving machine learning, formal proofs are critical for trust and reproducibility, especially when new protocol designs are introduced.

3. Limited Baseline Comparisons

The experimental section lacks comparisons with several state-of-the-art baselines that tackle similar privacy or security problems in federated learning. The authors primarily compare against a few limited methods, and newer or more sophisticated techniques, such as those incorporating homomorphic encryption, differential privacy, or secure multi-party computation (MPC) in a federated context, are not considered. This reduces the credibility of the performance evaluation and leaves the reader unsure how the method stands relative to the broader literature. I recommend authors to consider the following similar or close related works for in-depth comparison:

[1] Zhang C, Li S, Xia J, et al. {BatchCrypt}: Efficient homomorphic encryption for {Cross-Silo} federated learning[C]//2020 USENIX annual technical conference (USENIX ATC 20). 2020: 493-506.

[2] Liu J, Lou J, Xiong L, et al. Cross-silo federated learning with record-level personalized differential privacy[C]//Proceedings of the 2024 on ACM SIGSAC Conference on Computer and Communications Security. 2024: 303-317.

[3] So J, He C, Yang C S, et al. Lightsecagg: a lightweight and versatile design for secure aggregation in federated learning[J]. Proceedings of Machine Learning and Systems, 2022, 4: 694-720.

4. Narrow Application Scope

The proposed framework appears to be tailored for a relatively constrained federated learning setting. For example, the evaluation uses only generalized linear models and does not generalize convincingly to more diverse or large-scale real-world FL scenarios (e.g., deep learning-based settings). In addition, assumptions such as synchronous communication and ideal channel conditions may not hold in many real-world deployments (e.g., servers go offline). The lack of discussion of deployment limitations or scalability further limits the applicability of the approach.

**Strengths And Weaknesses:**

Strengths:
+ This work offers an MPC-based FL framework that trains differentially private linear models across data holders in cross-silo settings, supporting both horizontal and vertical data partitioning.
+ Extensive case studies.
+ The proposed method achieved first place in the prestigious iDASH competition.

Weaknesses:
- Lack of threat model introduction, such as attacker's goals, background knowledge, and capabilities.
- Lack of formal proofs related to privacy or security.
- More advanced baselines need to be added for further comparison.
- The application scenarios of the proposed method seem to be limited.

---

> ### Author Response · Authors · 2025-05-13
> **Response for Comments 1 and 2**
>
> Thank you for your comments and suggestions. We have carefully considered all feedback and provide detailed responses below.
>
> 1. The 2nd paragraph of Section 2.2 contains a description of the standard threat models used in the MPC literature (passive and active) which we adhere to. We also explicitly state the non-collusion assumption among MPC servers in footnote 6, which is a standard assumption in secure MPC protocols. In the same paragraph, we also mention that our MPC protocols are designed to be generic so that they work under different threat models. This is achieved by changing the underlying MPC scheme to align with the desired security setting. Our results in Table 6 and 7 demonstrate that our protocols function effectively under multiple adversarial settings (active and passive), therefore validating that our protocols are generic and can be used for a range of threat models.
>
>  We will make this more explicit in our results and in Section 2.2. Please let us know if you need more details and discussions around this in the paper.
>
> 2. Regarding input privacy, as mentioned in Section 2.2, our proposed protocols rely on standard MPC primitives, whose security is well-established under the Universal Composability (UC) framework. Regarding output privacy, we build on the result by Chaudhauri et al. (2011)
> who, as stated in Section 4.1, have shown that the output perturbation method provides $\epsilon$-DP as long as (C1) each input feature vector has an L2 norm of at most 1; and (C2) the LR model is trained using L2 regularization. For completeness, we will add the following subsection in the paper:
>
> **Security and Privacy of $\pi_{\mathsf{LR}} + \pi_{\mathsf{DP}}$**. Below we formalize the security guarantees of $\pi_{\mathsf{LR}} + \pi_{\mathsf{DP}}$.
>
>   Input privacy: The underlying MPC schemes that we use in our method implement a secure arithmetic black-box MPC. They only perform operations over secret shares, and no information is leaked during the computation over the secret shares. Moreover, we use MPC sub-protocols $\pi_{\mathsf{LOG}}$, $\pi_{\mathsf{COS}}$, $\pi_{\mathsf{SIN}}$, $\pi_{\mathsf{GR-RANDOM}}$, and $\pi_{\mathsf{LR}}$ from MP-SPDZ. All these sub-protocols do not leak any information and are UC-secure. The novel protocols that we propose ($\pi_{\mathsf{DP}}$, $\pi_{\mathsf{GSS}}$, and $\pi_{\mathsf{NORM}}$) therefore do not leak any information to the MPC servers or the data holders, except for the perturbed weights of the LR model. Our protocol $\pi_{\mathsf{LR}} + \pi_{\mathsf{DP}}$, which is a composition of $\pi_{\mathsf{LR}}$ and $\pi_{\mathsf{DP}}$, UC-securely implements the ideal functionality $\mathcal{F}_{\mathsf{LR+DP}}$ for privacy-preserving training of a differentially private logistic regression model.
>
> *Ideal Functionality $\mathcal{F}_{\mathsf{LR+DP}}$*: The functionality $\mathcal{F}_{\mathsf{LR+DP}}$ operates as follows:
> - Waits to receive input datasets $D_i$ from data holders and joins them to form $D$.
> - Upon joining $D$, computes model weights $\mathbf{w} := \mathcal{A}(D)$ by training a logistic regression model $\mathcal{A}$ on $D$.
> - Samples a noise vector $\mathbf{s} \sim p(\eta) \propto h(\eta)$.
> - Computes the privatized output $\tilde{\mathbf{w}} := \mathbf{w} + \mathbf{s}$.
> - Reveals $\tilde{\mathbf{w}}$.
>
> Output privacy: The fact that the resulting $\tilde{\mathbf{w}}$ provides $(\epsilon,0)$-DP follows directly from the proof by Chaudhauri et al. (2011) regarding the privacy guarantees of the output perturbation method. Although in our approach the training and noise addition procedures are executed within MPC using fixed-point representations (in our case, precision of 32-bit), prior work by Mironov (2012) has shown that DP mechanisms remain valid under finite precision, even at 32 bits. Therefore, our protocol $\pi_{\mathsf{LR+DP}}$, securely computes $\tilde{\mathbf{w}}$ and satisfies the same $(\epsilon, 0)$-DP guarantees.

---

> ### Author Response · Authors · 2025-05-13
> **Response for Comments 3 and 4**
>
> 3. Thank you for providing the references; we will add them to our related work.
>
>     The methods proposed in [1] and [3] only provide input privacy and are therefore prone to membership inference attacks [Section 7.4 in (Zhao et al., 2025)]. As stated in Section 3.1 we do not include methods that do not privacy DP guarantees in our empirical comparison. The lack of output privacy protection makes these methods -- including those in [1] and [3] -- much weaker from a privacy perspective than all the methods that we empirically evaluate.
>
>    Reference [2] does provide both input and output privacy, but it does so using local DP (using DP-SGD on the client side), which leads to lower utility than global DP. Row 4 in Table 4 provides an upper estimate of the accuracy that can be achieved in this way. The technique in reference [2] does additional client- and record-level sampling during training, which leads to further accuracy loss.
>
> 4. While adopting MPC+DP is general and offers several advantages (as discussed in Section 3.3), it requires the design and development of custom MPC protocols tailored to each specific algorithm. Designing MPC protocols for every possible algorithm is beyond the scope of a single paper, however, our work serves as a foundation for future extensions to more complex models. We demonstrate the feasibility and effectiveness of MPC+DP by developing MPC protocols for output perturbation for GLMs, which, to our knowledge, has not been previously explored using MPC (please see Section 3.3, particularly in contrast to the recent work by Das et al. regarding MPC for deep learning).
>
>    An MPC+DP method for models like logistic regression, as we developed, is extremely valuable. First, while deep learning is state-of-the-art for many applications, GLMs remain a preferred method in domains such as finance and healthcare, where  interpretability and regulatory compliance are essential. Second, logistic regression is often more practical and effective than deep learning in real world problems with data scarcity (e.g. training of AI models for rare diseases). The effectiveness of GLMs is supported, for instance, by our winning results on real-world datasets from the iDASH competition (Table 1) and Bailly et al. (2022). Third, and very relevant to our work, is that GLMs are significantly more efficient than deep learning. While it is technically possible to train larger neural networks in MPC (Keller et al.), the computation and communication costs when doing so over encrypted data are orders of magnitude higher than for logistic regression. While there are applications where such high costs are justifiable to achieve higher accuracy, there are also many applications where logistic regression performs at par with neural networks, and training a logistic regression model in MPC is a far more economical and practical solution.
>
>
> Reference:
> - Zhao, Joshua, et al. "The Federation Strikes Back: A Survey of Federated Learning Privacy Attacks, Defenses, Applications, and Policy Landscape." ACM Computing Surveys 57.9 (2025).
> - Bailly, Alexandre, et al. "Effects of dataset size and interactions on the prediction
> performance of logistic regression and deep learning models." Computer Methods and Programs in
> Biomedicine 213 (2022): 106504.
> - Keller, Marcel, and Ke Sun. "Secure quantized training for deep learning." International Conference on Machine Learning. PMLR, 2022.

---

### Decision · Action_Editor_So8U · 2025-06-17

**Recommendation:** Reject

**Additional Comments:**

Main concerns with the paper, also mentioned by the reviewers, are the lack of formal proofs for the claims made and insufficient clarity regarding the contributions.

As reviewer g5C2 points out, there is "Lack of formal proofs related to privacy or security."  I also could not find rigorous results supporting the privacy guarantees. Related to DP and output perturbation, for example, there is no formal explanation why normalization of the features is carried out. It would be good to formally state all the required results.

The proposed method appears to be an application of known techniques in the context of differentially private training using MPC. While the authors mention that this setting is for the "iDASH2021 Track III competition on confidential computing, where the challenge was to propose a federated learning algorithm for training of a model to predict the risk of wild-type transthyretin amyloid cardiomyopathy using medical claims data from different hospitals, while providing DP guarantees", the technical contributions are not made sufficiently clear. Overall, the manuscript would benefit from a more precise and explicit discussion of what is being claimed as a contribution and how it builds on prior work.

In case of resubmission to TMLR, I would recommend going carefully through all reviewer comments, providing the required formal results to support claims of privacy and security and clearly stating the intended contributions of the paper.

**Audience:**

Yes

**Audience Explanation:**

I think the topic is interesting to a large part of TMLR audience: federated learning with DP combined with secure multi-party computation.

**Claims And Evidence:**

No

**Claims Explanation:**

This YES/NO is a borderline. There is an extensive experimental section, comparing the computational efficiency of the proposed scheme with the baselines and the privacy-utilty tradeoffs of the proposed output perturbation scheme with baselines. On the other hand, some of the results are confusing: e.g. Table 3 has two different results for the same value of $\epsilon$ and the accuracy drops as $\epsilon$ increases. Also, I find it slightly suspicious that DP-SGD is so much worse than output perturbation (Table 4). The view of reviewers Cbds and bUUn is also that more comparisons are needed, with stronger baselines and more difficult datasets.

**Resubmission Of Major Revision:**

The authors may consider submitting a major revision at a later time.